# Trodusquemine enhances Aβ$_{42}$ aggregation but suppresses its toxicity by displacing oligomers from cell membranes

Ryan Limbocker[1], Sean Chia[1], Francesco S. Ruggeri[1], Michele Perni [1], Roberta Cascella[2], Gabriella T. Heller[1], Georg Meisl [1], Benedetta Mannini [1], Johnny Habchi[1], Thomas C.T. Michaels[1,3], Pavan K. Challa[1], Minkoo Ahn [1], Samuel T. Casford[1], Nilumi Fernando[1], Catherine K. Xu[1], Nina D. Kloss[1], Samuel I.A. Cohen[1], Janet R. Kumita [1], Cristina Cecchi[2], Michael Zasloff[4], Sara Linse [5], Tuomas P.J. Knowles[1,6], Fabrizio Chiti [2], Michele Vendruscolo[1] & Christopher M. Dobson [1]

Transient oligomeric species formed during the aggregation process of the 42-residue form of the amyloid-β peptide (Aβ$_{42}$) are key pathogenic agents in Alzheimer's disease (AD). To investigate the relationship between Aβ$_{42}$ aggregation and its cytotoxicity and the influence of a potential drug on both phenomena, we have studied the effects of trodusquemine. This aminosterol enhances the rate of aggregation by promoting monomer-dependent secondary nucleation, but significantly reduces the toxicity of the resulting oligomers to neuroblastoma cells by inhibiting their binding to the cellular membranes. When administered to a *C. elegans* model of AD, we again observe an increase in aggregate formation alongside the suppression of Aβ$_{42}$-induced toxicity. In addition to oligomer displacement, the reduced toxicity could also point towards an increased rate of conversion of oligomers to less toxic fibrils. The ability of a small molecule to reduce the toxicity of oligomeric species represents a potential therapeutic strategy against AD.

[1] Centre for Misfolding Diseases, Department of Chemistry, University of Cambridge, Cambridge CB2 1EW, UK. [2] Department of Experimental and Clinical Biomedical Science, University of Florence, Florence 50134, Italy. [3] Paulson School of Engineering and Applied Sciences, Harvard University, Cambridge, MA 02138, USA. [4] MedStar-Georgetown Transplant Institute, Georgetown University School of Medicine, Washington, DC 20010, USA. [5] Department of Biochemistry and Structural Biology, Lund University, SE221 00 Lund, Sweden. [6] Cavendish Laboratory, Department of Physics, University of Cambridge, Cambridge CB3 0HE, UK. Correspondence and requests for materials should be addressed to F.C. (email: fabrizio.chiti@unifi.it) or to M.V. (email: mv245@cam.ac.uk) or to C.M.D. (email: cmd44@cam.ac.uk)

Alzheimer's disease (AD) is a fatal neurodegenerative disorder characterized by aberrant protein aggregation, which results in multifactorial neuronal dysfunction affecting synaptic signaling, mitochondrial function, neuroinflammation and neuronal loss[1–6]. Although the pathophysiology of AD is extremely complex and heterogeneous, the buildup of amyloid plaques in the extracellular space of the brain parenchyma is a hallmark of this disease. These proteinaceous deposits form as a consequence of the aggregation of the intrinsically disordered amyloid-β peptide (Aβ), a proteolytically derived transmembrane fragment of the amyloid precursor protein (APP), and a multitude of biochemical, genetic and animal investigations point to the aberrant behaviour of this molecule as central to the aetiology of AD[1–5,7].

A range of small molecules and antibodies have been characterized for their ability to modulate the self-assembly process of the Aβ peptide[8–12]. So far, however, no clinical trial based on such compounds has been successful[13]. This situation can be attributed in part to the limited understanding of the mechanisms by which aggregation occurs and of the means by which these compounds modify the aggregation process, and also to their administration at too late a stage in a clinical situation where the amyloid load has already reached significant levels[9]. To address this challenge, we have developed a rational drug discovery strategy based on chemical kinetics to elucidate, in molecular detail, the effects of candidate compounds on the microscopic processes, in particular primary and secondary nucleation and elongation, that underlie the aggregation phenomenon[8,9,14]. The 42-residue form of the Aβ peptide (Aβ42) is a primary component of the amyloid deposits in AD, and we have found that both small and large molecules can alter, often dramatically, one or more of these specific steps[8,9,15,16].

Increasing evidence indicates that oligomeric species of Aβ42 formed as intermediates during the aggregation process are substantially more toxic to neuronal cells than are the mature fibrils or plaques, and therefore are likely to contribute very significantly to the onset and spread of disease[5,7,17]. In addition, it has been shown that molecular chaperones and antibodies can reduce the levels or degree of toxicity of such oligomeric intermediates[15,18,19]. Therefore, other molecules with similar behaviour could be efficacious as therapeutic agents to combat AD by targeting the most deleterious species associated with Aβ42 aggregation. In the present study, we investigate the effects of the aminosterol trodusquemine (also known as MSI-1436), a natural product that was first isolated from the liver of the dogfish shark[20], on the aggregation of Aβ42.

Trodusquemine belongs to a family of compounds shown to displace proteins from membranes, a feature proposed to preserve membrane integrity[21]. Indeed, trodusquemine and the closely related aminosterol squalamine have been shown to inhibit α-synuclein aggregation and its related toxicity by displacing both monomers and oligomers from the membranes of cultured neuroblastoma cells, and also to suppress the onset of paralysis in a *C. elegans* model of Parkinson's disease[22,23]. As drug candidates, aminosterols have a well-characterized pharmacokinetic activity, and high tolerability and safety profiles in humans, as a result of several past and ongoing clinical trials, and at least one (trodusquemine) has been reported to be able to cross the blood–brain barrier[24–26]. We show that despite enhancing the rate of Aβ42 aggregation in vitro by accelerating the microscopic step of secondary nucleation, trodusquemine ameliorates the toxicity of Aβ42 oligomers in neuroblastoma cells by reducing their binding affinity to cellular membranes, and also decreases the toxicity inherent to Aβ42 aggregation in a *C. elegans* model of AD when administered before or during the manifestation of a toxic phenotype.

## Results

**Effects of trodusquemine on Aβ42 aggregation.** Trodusquemine consists of a fused sterol ring covalently bound to the polyamine spermine (Fig. 1a)[25]. Polyamines are small cationic molecules that have been found to increase the rate of Aβ40 aggregation in vitro[27]; although, the mechanism by which this process takes place has yet to be elucidated. Initially, we followed Aβ42 fibril formation in vitro using the well-defined thioflavin-T (ThT)-based chemical kinetics assay[28]. We monitored the aggregation of monomeric Aβ42 at a concentration of 2 μM in 20 mM sodium phosphate, 200 μM EDTA, pH 8.0, 37 °C, under quiescent conditions, in the absence and presence of trodusquemine at ratios of Aβ42-to-trodusquemine of 10:1, 5:1 and 1:1 molar equivalents (Fig. 1b). The results revealed that in the presence of this compound, the half-time of Aβ42 aggregation was reduced in a dose-dependent manner, and by a factor of two in the presence of a 5:1 ratio of Aβ42-to-trodusquemine. A similar, dose-dependent, accelerating effect has been observed with poly-L-lysine, in which case a factor of two reduction in the half-time of Aβ42 aggregation was observed at a 1:10 molar ratio of Aβ42-to-lysine[29].

To probe the mechanism of association between trodusquemine and Aβ42 and to assess the importance of the side chain of trodusquemine in the promotion of fibril formation, we next monitored the aggregation of Aβ42 in the presence of spermine. We observed that this cationic polyamine accelerates the Aβ42 aggregation reaction in a dose-dependent manner (Supplementary Figure 1); however, the covalent linkage of spermine to the sterol ring appears to increase the efficacy of trodusquemine above that of spermine alone. The accelerative effects of spermine and trodusquemine were further investigated at physiological ionic strength (5 mM sodium phosphate, 150 mM NaCl, 200 μM EDTA, pH 8.0) where electrostatic interactions between opposing charges are weakened[30]. We observed that the ability of spermine to enhance the aggregation reaction was diminished with increasing concentrations of salt. Conversely, the accelerative effects of trodusquemine were maintained at physiological ionic strength (Supplementary Figure 1). Additional factors other than electrostatic interactions between the spermine moiety and negatively charged residues of Aβ42 are therefore likely to contribute to the acceleration of aggregation by trodusquemine; for example, hydrophobic interactions made possible by its sterol ring may provide a molecular basis for its increased potency in comparison to spermine alone.

Next, we assessed the effects of trodusquemine on secondary processes by monitoring the aggregation kinetics of Aβ42 in the presence of pre-formed fibrils, which act as seeds to promote aggregate growth and multiplication. The aggregation of Aβ42 under these conditions was found, as in previous experiments, to experience an efficient positive feedback loop as fibril surfaces act as catalytic sites for secondary nucleation. This phenomenon, therefore, gives rise to greatly enhanced conversion of soluble monomeric Aβ42 into oligomeric species that are then able to elongate to form new fibrils, as surface induced secondary nucleation has been shown previously to dominate the in vitro aggregation reaction of Aβ42 once primary nucleation has generated a low but critical concentration of the total Aβ42 aggregates[31]. The addition of seed fibrils at the beginning of the aggregation reaction, therefore, is able to increase the rate of fibril formation by promoting secondary nucleation (with rate constant $k_2$) and elongation (with rate constant $k_+$); under these conditions, primary nucleation (with rate constant $k_n$) is no longer rate-limiting and of negligible significance in driving the rate of Aβ42 fibril formation[8,31,32]. We followed the aggregation of 2 μM monomeric Aβ42 in the presence of 5% seed fibrils (monomer equivalents, prepared as previously described[9]) and trodusquemine at ratios of Aβ42-to-trodusquemine of 10:1, 5:1

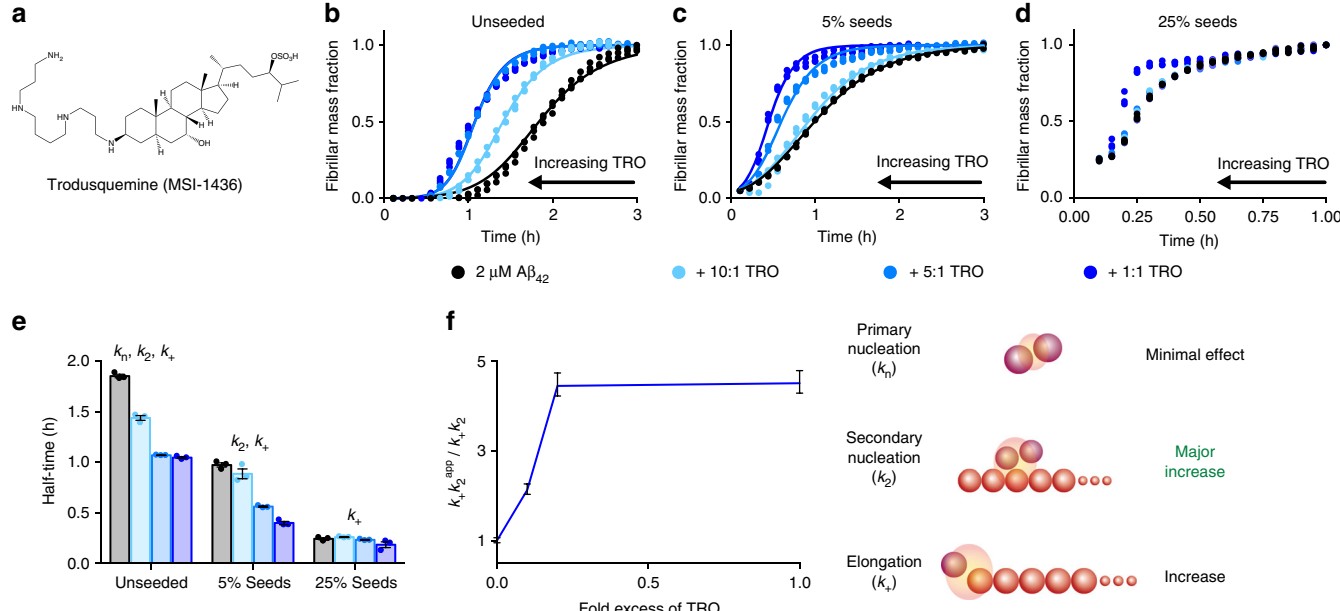

**Fig. 1** Trodusquemine enhances the overall rate of Aβ$_{42}$ aggregation by promoting secondary nucleation. **a** Structure of trodusquemine (TRO). Kinetic profiles of the aggregation of 2 μM Aβ$_{42}$ in the absence (black) or presence of trodusquemine at ratios of Aβ$_{42}$-to-trodusquemine of 10:1, 5:1 and 1:1 (light to dark blue) in the absence of seeds (**b**), in the presence of 5% seeds (**c**) and in the presence of 25% seeds (**d**). Solid lines indicate theoretical predictions based on kinetic fitting (see Results); the data are very well-described by varying the rates associated with the presence of trodusquemine in the secondary pathway ($k_+k_2$) (**b**), or associated with monomer-dependent secondary nucleation ($k_2$) (**c**). **e** Change of the half-time of aggregation of Aβ$_{42}$ as seen from **b** to **d**. **f** Dependency of the apparent reaction rate constants of secondary pathways ($k_+k_2^{app}$) on increasing concentrations of trodusquemine relative to the condition in the absence of the molecule ($k_+k_2$). The schematic depicts the microscopic steps involved in Aβ$_{42}$ aggregation. In all panels, error bars indicate standard error of the mean (s.e.m.) of three replicates

and 1:1 (Fig. 1c). As observed in the unseeded reactions, the half-time of Aβ$_{42}$ aggregation decreased with increasing concentrations of trodusquemine in a well-defined dose-dependent manner. As primary nucleation is not rate-limiting in this seeded aggregation assay, these results indicate that the enhanced aggregation of Aβ$_{42}$ in the presence of trodusquemine is not associated with changes in the primary nucleation step.

Additional experiments were carried out in the presence of higher levels of seed fibrils, where elongation of the pre-formed fibrils has been found to be the dominant process of aggregate growth; under these conditions, neither primary nor secondary nucleation processes are rate-limiting[31]. We found that the rate of aggregation of 2 μM monomeric Aβ$_{42}$ in the presence of 25% of seed fibrils and of trodusquemine at ratios of Aβ$_{42}$-to-trodusquemine of 10:1, 5:1 and 1:1 was increased only at the highest ratio (i.e. 1 molar equivalent) of the compound to Aβ$_{42}$, and then only slightly (the half-time was reduced from 0.28 to 0.19 h) (Fig. 1d), confirming that trodusquemine exerts only a minimal effect on the rate of Aβ$_{42}$ elongation. Collectively, the unseeded and seeded macroscopic aggregation experiments (Fig. 1e), therefore, strongly suggest that the enhanced aggregation of Aβ$_{42}$ in the presence of trodusquemine is the result of an increased rate of surface-catalyzed secondary nucleation.

We then carried out a quantitative analysis of the effects of trodusquemine on Aβ$_{42}$ aggregation by comparing the experimental kinetic data from the unseeded and seeded aggregation experiments with kinetic curves obtained by solving analytically the rate laws of aggregation kinetics, as described in detail previously[31,33,34]. This methodology enables the macroscopic time-dependent proliferation of aggregates to be expressed as a function of the rate constants for each of the contributing microscopic processes. In the presence of trodusquemine, the

aggregation profiles were analyzed by introducing perturbations of the microscopic constants into the rate laws, in order to identify the effects on the specific processes that are affected[9,14]. Using this approach, we observed that the unseeded kinetic data in the presence of different concentrations of trodusquemine (Fig. 1b) were very well-described by an increase in the product of the rate constants involving secondary pathways ($k_+k_2$) (Fig. 1f); moreover, the seeded aggregation data (Fig. 1c) were found to fit closely to a situation where the observed increase in rate constants can be attributed to an increase in the rate of monomer-dependent secondary nucleation. By contrast, analyzing the data for an effect on the rates of primary nucleation failed to recapitulate the observed effects of trodusquemine on the aggregation reaction (see Supplementary Note 1, Supplementary Figure 2).

We also examined the interaction of trodusquemine with monomeric Aβ$_{42}$ using $^1$H-$^{15}$N-HSQC nuclear magnetic resonance (NMR) spectroscopy at 5 °C with 50 μM uniformly $^{15}$N-labeled Aβ$_{42}$ and 50 μM trodusquemine in 5 mM sodium phosphate at pH 7.4. The presence of trodusquemine did not significantly perturb the observed amide cross-peak positions or intensities (Supplementary Figure 3, Supplementary Table 1), indicating that its binding to monomeric Aβ$_{42}$ in the aggregation reaction is very weak at the molar ratios investigated in the kinetic experiments. Nonetheless, it is possible that even weak interactions of trodusquemine with monomeric Aβ$_{42}$ could potentially promote its binding to fibrils, or that trodusquemine may stabilize oligomeric species; either of these situations could be related to the observed enhancement in the rate of nucleation by trodusquemine. Taken together, therefore, all these experiments indicate that trodusquemine is able to enhance significantly the rate of secondary nucleation of Aβ$_{42}$.

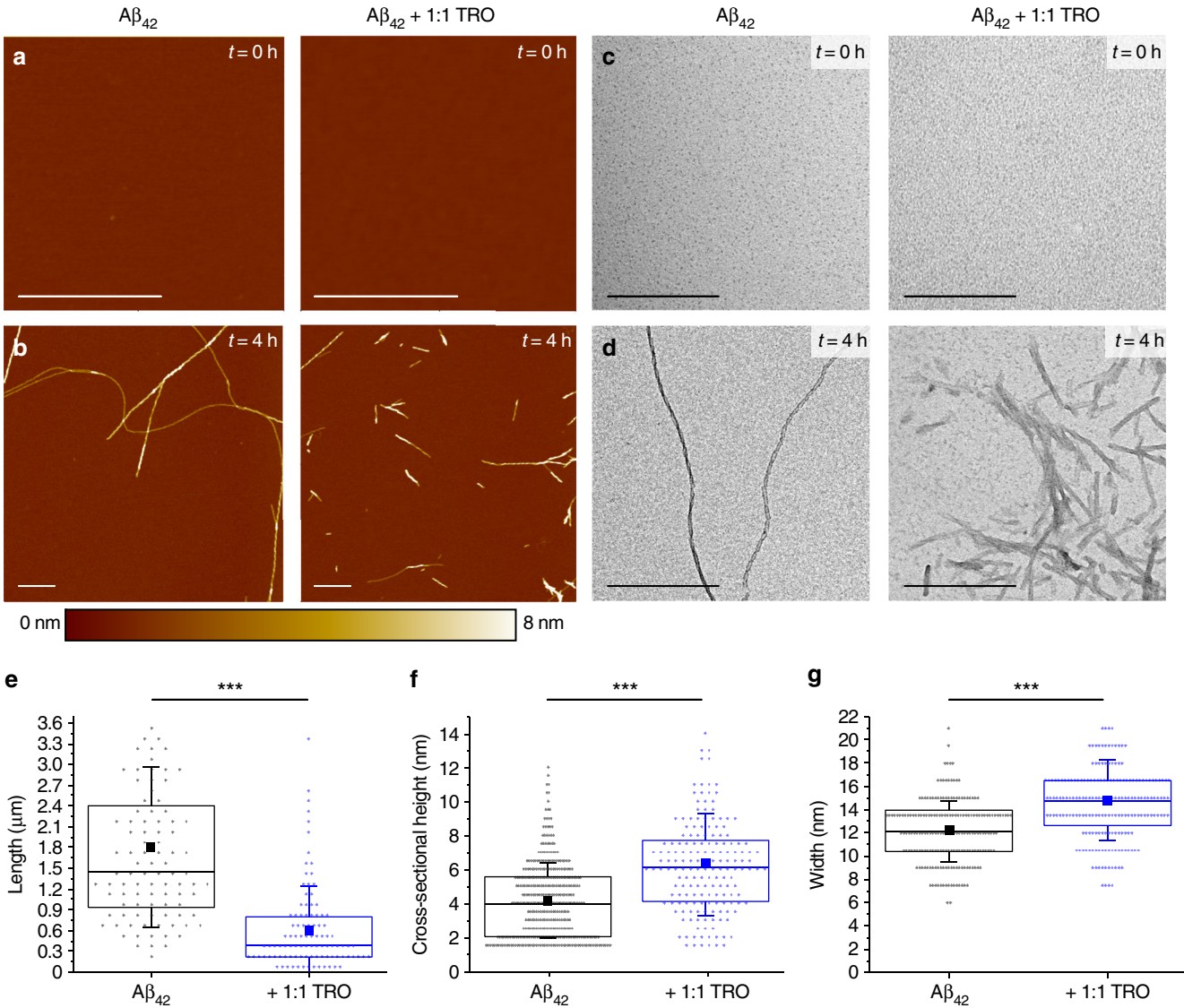

**Fig. 2** Trodusquemine promotes the formation of shorter and thicker Aβ$_{42}$ fibrils in vitro. AFM (**a**, **b**) and TEM (**c**, **d**) measurements of Aβ$_{42}$ in the absence (left panels) or presence (right panels) of an equimolar concentration of trodusquemine. Samples were prepared at the initiation stage ($t = 0$ h, **a**, **c**) and in the plateau region ($t = 4$ h, **b**, **d**) of the aggregation reaction. Scale bars, 500 nm. AFM distributions of fibril lengths (**e**) ($N = 100$ per condition, ***$P <$ 0.001 by paired $t$-test) and cross-sectional heights (**f**) ($N = 387$ for Aβ$_{42}$ and $N = 174$ for Aβ$_{42}$ + 1:1 TRO, ***$P < 0.001$ by paired $t$-test), and TEM widths (**g**) ($N = 200$ per condition, ***$P < 0.001$ by paired $t$-test) in the absence (black) and presence (blue) of trodusquemine. In **e**–**g** the square denotes the mean, the centre line indicates the median, the bounds of the boxes indicate the first and third quartiles containing 50% of the data, and the whiskers indicate the first standard deviation

**Statistical analysis of Aβ$_{42}$ fibril size and morphology.** In order to examine independently the conclusions drawn from the experiments using chemical kinetics and to understand in more detail the observed increase in the ThT fluorescence intensity with increasing concentrations of trodusquemine (Supplementary Figure 4), we carried out a statistical analysis of fibril morphology by means of high-resolution and phase controlled atomic force microscopy (AFM)[35,36] and transmission electron microscopy (TEM). Aliquots of the different samples were deposited, either on atomically flat and positively functionalized mica surfaces for AFM or on carbon coated copper grids for TEM, at the initiation stage ($t = 0$ h) and in the plateau region ($t = 4$ h) of the aggregation reaction of monomeric Aβ$_{42}$ at a concentration of 2 μM in the absence and presence of an equimolar ratio of Aβ$_{42}$-to-trodusquemine (Fig. 2a–d, Supplementary Figure 5).

We first analyzed the lengths (Fig. 2e) and cross-sectional heights (Fig. 2f, Supplementary Figure 6) of the individual fibrils that were measured by means of high-resolution and phase controlled AFM, which facilitates a precise morphological comparison of aggregates across samples and minimizes the effects of sample deformation (<10%) by maintaining a regime of low tip-sample interaction (see Methods for details)[35,36]. We then measured the widths of the individual fibrils that were imaged by TEM (Fig. 2g), which, in combination with the three-dimensional AFM maps, enabled us to understand in greater detail the structural properties of the fibrils. Incubation of Aβ$_{42}$ in the absence of trodusquemine produced fibrils with an average height of 4.2 ± 0.2 nm (mean ± total error, see Methods for details; $n =$ 387), a width of 12 ± 1 nm ($n = 200$) and a length of 1.81 ± 0.12 μm ($n = 100$), while in the presence of an equimolar ratio of

$A\beta_{42}$-to-trodusquemine, the fibrils had an average height of $6.3 \pm 0.3$ nm ($n = 174$), a width of $15 \pm 1$ nm ($n = 200$) and a length of $0.63 \pm 0.06$ μm ($n = 100$) (Fig. 2e–g).

Both techniques indicate that the incubation of $A\beta_{42}$ in the presence of trodusquemine resulted in the formation of fibrils characterized by larger cross-sectional dimensions, as observed by the increased average height (Fig. 2f) and width (Fig. 2g), in comparison to fibrils formed in the absence of the molecule (see Supplementary Note 2 for an AFM and TEM size comparison). The comparison of the length distributions of the two types of fibrillar aggregates measured with AFM showed that fibrils formed in the presence of trodusquemine were markedly shorter than fibrils formed in their absence (Fig. 2e); this phenomenon was particularly evident in the TEM images (Fig. 2d). An increase in the frequency of nucleation events is expected to result in the formation of shorter fibrils. Indeed, in a system dominated by secondary nucleation, the average fibril length is expected to scale with[37]

$$\sqrt{(k_+/k_2)}. \tag{1}$$

Hence, the measured increase of the rate constant for secondary nucleation, $k_2$, by a factor of 5 in the presence of equimolar concentrations of $A\beta_{42}$-to-trodusquemine (Fig. 1f) is predicted to result in a decrease of the average fibril length by a factor of $1/\sqrt{(5)} \approx 0.5$. This prediction is consistent with the experimentally observed average length that is reduced by a factor of $\approx 0.4$ (Fig. 2e), adding further support to the conclusions of the analysis of the chemical kinetics experiments that surface-catalyzed secondary nucleation is enhanced by trodusquemine. Moreover, TEM measurements after 1 h of aggregation demonstrated that the presence of trodusquemine led to the production of numerous epicenters of increased nucleation, as evidenced by the significantly increased formation of localized short fibrils in groups (see Supplementary Note 3, Supplementary Figure 7). Collectively, therefore, all the results reported in this paper demonstrate that trodusquemine utilizes the surface of pre-formed fibrils of $A\beta_{42}$ to nucleate rapidly via secondary nucleation to generate a multitude of short fibrils.

**Trodusquemine reduces $A\beta_{42}$ oligomer toxicity to cells**. The $A\beta$ oligomers that form during an ongoing aggregation process reach a maximum of ca. 1% of the total monomer concentration at the point in time where ca. half the monomers are converted to fibrils[31]. Many studies have therefore used methods to produce and stabilize oligomers formed at higher concentrations in order to make it possible to study the properties of such species in detail[38,39]. Adhering to such practice, we next measured the effects of trodusquemine on the toxicity of $A\beta_{42}$ oligomers by means of $A\beta$-derived diffusible ligands (ADDLs). These small and soluble aggregates (referred to hereafter as $A\beta_{42}$ oligomers) are a frequently employed oligomeric model and have been found to accumulate in the brains of Alzheimer's disease patients, as shown by conformation-sensitive antibodies raised against them[40], and are capable of causing neurological damage, attenuating mitochondrial activity, increasing membrane permeability, promoting the production of reactive oxygen species (ROS), inhibiting long-term potentiation (LTP) and causing abnormal synapse composition, shape, and density[40–44].

$A\beta_{42}$ oligomers were generated as previously described[41] from both recombinant and synthetic peptides (see Supplementary Note 4) and added to the cell culture media of human neuroblastoma SH-SY5Y cell lines at the same concentration (monomer equivalents). Following the observation that the two

oligomer forms reduced the viability of the cells to similar extents using the 3-(4,5-dimethylthiazol-2-yl)-2,5-diphenyltetrazolium bromide (MTT) reduction assay (Supplementary Figure 8), we decided to use the synthetic $A\beta_{42}$ oligomers for further studies. The structural characteristics of these $A\beta_{42}$ oligomers was investigated by western blot and dot blot assay using a conformation-sensitive anti-ADDLs antibody (Supplementary Figure 9) and found to be in good agreement with previous reports[41].

The $A\beta_{42}$ oligomers were added to cell culture media at a concentration of 1 μM (monomer equivalents) and the viability of neuroblastoma SH-SY5Y cells was found to be reduced by $24 \pm 3\%$ (all cell biology results are described by mean ± s.e.m. of three-independent experiments), as measured by their ability to reduce MTT (Fig. 3a). The $A\beta_{42}$ oligomers were then incubated for 1 h at 37 °C at a concentration of 1 μM (monomer equivalents) in the absence or presence of different concentrations of trodusquemine (from 0.1 to 1.0 μM) and subsequently added to the cell culture medium for 24 h. Trodusquemine was found to decrease the toxicity of the oligomers significantly, such that at a 1:1 ratio of $A\beta_{42}$-to-trodusquemine, cell viability was increased by $22 \pm 2\%$, up to $96 \pm 2\%$ relative to untreated cells (Fig. 3a), a value approximately equal to that of cells treated with 1 μM trodusquemine in the absence of oligomers ($97 \pm 1\%$ of untreated cells). The $A\beta_{42}$ oligomers also caused a significant increase in ROS levels in SH-SY5Y cells ($328 \pm 14\%$ of the value for untreated cells) (Fig. 3b), as monitored by the fluorescent probe 2′,7′-dichlorodihydrofluor-escein diacetate (CM-H$_2$DCFDA), in agreement with previous observations[43]. The level of ROS-derived fluorescence was observed to decrease substantially when the oligomers were co-incubated with 1 μM trodusquemine ($163 \pm 15\%$), to a level similar to that of cells treated with cell culture medium ($100 \pm 12\%$) or 1 μM trodusquemine alone ($119 \pm 6\%$, Fig. 3b).

To explore further the mechanism of the reduction of oligomer-induced toxicity by trodusquemine, we monitored the interaction of the $A\beta_{42}$ oligomers with SH-SY5Y cellular membranes in the absence and in the presence of trodusquemine using confocal microscopy. The cells were incubated with oligomers (1 μM monomer equivalents) for 15 min in the absence or presence of 1 μM trodusquemine and images were scanned at the apical planes. The oligomers and the membranes were visualized by staining with the sequence-specific 6E10 anti-$A\beta$ antibody followed by the Alexa Fluor 488-conjugated secondary antibody (green channel) and the Alexa Fluor 633-conjugated wheat germ agglutinin (red channel), respectively. Large numbers of oligomers were observed to be bound to the cell membranes in the absence of trodusquemine[45], whereas in its presence the extent of binding to the cellular membranes was reduced by more than 75% (Fig. 3c). As previous work has shown that toxicity monitored by the MTT reduction assay and ROS production correlates directly with the extent of the interaction of the oligomers with cell membranes[22,46], this finding is consistent with the greatly reduced toxicity of $A\beta_{42}$ aggregates in the presence of trodusquemine.

As a control, we monitored the co-localization of $A\beta_{42}$ oligomers with lysosomes using HiLyte Fluor 647-labeled $A\beta_{42}$ and the LysoTracker probe (Supplementary Figure 10). We observed negligible activation of lysosomes in cells exposed to oligomers and minimal co-localization of the $A\beta_{42}$ oligomers with lysosomes after 15 min of treatment. To ensure further that the oligomers observed were bound to the membranes (Fig. 3c) and not lysosomal, we also analyzed the median planes of the cells (Supplementary Figure 11). Intracellular green fluorescence resulting from the oligomers was not observed in these planes, indicating strongly that the total fluorescence observed in all planes arises only from the oligomers bound to the membranes.

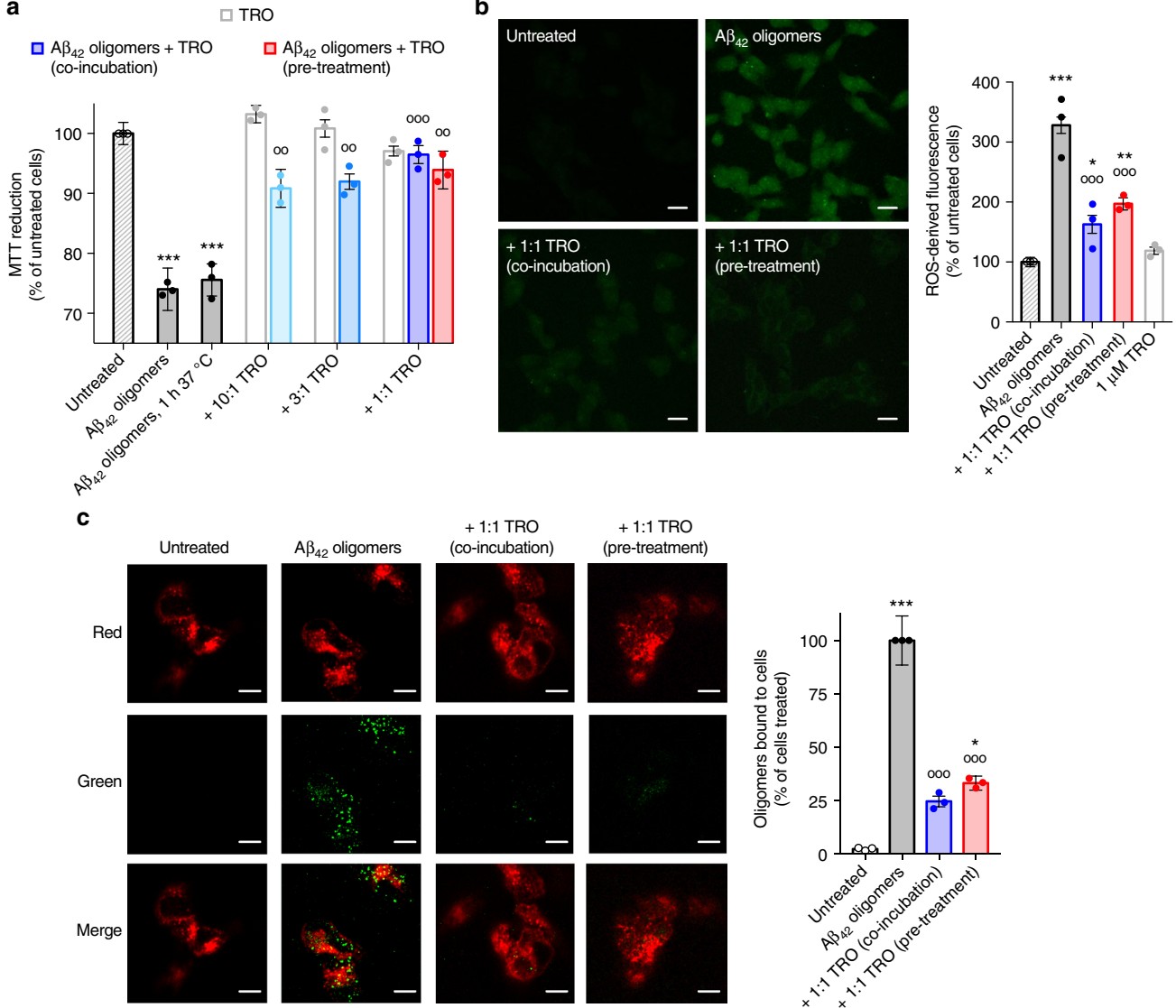

**Fig. 3** The toxicity of Aβ₄₂ oligomers is reduced by trodusquemine. **a** Effects of trodusquemine on the Aβ₄₂ oligomer-induced decrease of MTT reduction. Oligomers (1 μM) were incubated in the absence or presence of 10:1–1:1 ratios of Aβ₄₂-to-trodusquemine (blue bars) for 1 h at 37 °C and incubated with cells for 24 h. Cells were also treated with corresponding concentrations of pre-incubated trodusquemine (white bars), or pre-treated with 1 μM trodusquemine, washed and exposed to oligomers under the same conditions (red bar). Samples containing oligomers were analyzed by one-way ANOVA followed by Bonferroni's multiple comparison test relative to untreated cells (***$P < 0.001$) and cells treated with oligomers (open circles, °°$P < 0.01$, °°°$P < 0.001$). A total of 80,000–100,000 cells were analyzed per condition in total. **b** Effects of trodusquemine on Aβ₄₂ oligomer-induced ROS production. Oligomers (1 μM) were incubated with cells for 15 min in the absence or presence of 1 μM trodusquemine. Cells were also treated with 1 μM trodusquemine. The green fluorescence arises from the CM-H₂DCFDA probe. The corresponding semi-quantitative values of the green fluorescence signals are shown. The red bar indicates the conditions shown in **a**. Scale bars, 30 μm. Samples containing oligomers were analyzed by one-way ANOVA followed by Bonferroni's multiple comparison test relative to untreated cells (*$P < 0.05$, **$P < 0.01$, ***$P < 0.001$) and cells treated with oligomers (°°°$P < 0.001$). A total of 100–120 cells were analyzed per condition in total. **c** Representative confocal scanning microscopy images of the apical sections of cells treated for 15 min with oligomers (1 μM) in the absence or presence of 1 μM trodusquemine. Red and green fluorescence indicate the cell membranes and the oligomers, respectively. The histogram shows the percentage of co-localization. The red bar indicates the conditions shown in **a**. Scale bars, 10 μm. Samples containing oligomers were analyzed by one-way ANOVA followed by Bonferroni's multiple comparison test relative to untreated cells (*$P < 0.05$, ***$P < 0.001$) and cells treated with oligomers (°°°$P < 0.001$). A total of 50–60 cells were analyzed per condition in total. In all panels, data represent mean ± s.e.m. of three independent experiments

In an additional set of experiments, we monitored the effects of trodusquemine when administered to cells as a pre-treatment prior to their exposure to toxic oligomers formed in the absence of the molecule. Cells were incubated with trodusquemine for 15 min and subsequently washed to remove any unbound molecules. Aβ₄₂ oligomers were then administered at an equimolar concentration under the conditions described previously above, and a marked decrease in the toxicity of the oligomers was observed, as monitored by cell viability (94 ± 3% of untreated cells, Fig. 3a) and ROS production (197 ± 10% of untreated cells, Fig. 3b). A prominent decrease in oligomer binding to the cellular membrane was also observed (33 ± 3% of treated cells, Fig. 3c). Indeed, the MTT reduction, ROS levels and oligomer binding measurements for pre-treated cells are highly similar to those

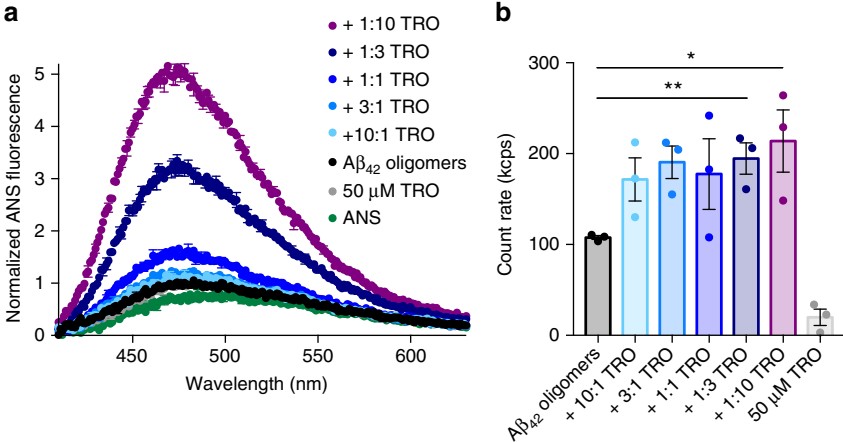

**Fig. 4** Trodusquemine modifies the biophysical properties of $A\beta_{42}$ oligomers. **a** The binding of 15 μM ANS to 5 μM $A\beta_{42}$ oligomers after incubation for 1 h at 20 °C in the absence (black) and presence of 10:1, 3:1, 1:1, 1:3 and 1:10 ratios (light blue to purple) of $A\beta_{42}$-to-trodusquemine. Spectra were normalized to the intensity of ANS fluorescence for $A\beta_{42}$ oligomers. Mean ± s.e.m. of duplicate samples. Data shown are representative of three-independent experiments that yielded similar results. **b** Static light scattering measurements of 5 μM $A\beta_{42}$ oligomers after incubation for 1 h at 20 °C in the absence or presence of 10:1, 3:1, 1:1, 1:3 and 1:10 ratios of $A\beta_{42}$-to-trodusquemine. Mean ± s.e.m. of three replicates. *$P < 0.05$, **$P < 0.01$ by Student's $t$-test

described above for cells co-incubated with oligomers and trodusquemine.

**Trodusquemine modifies the properties of $A\beta_{42}$ oligomers.** Several studies have demonstrated that the extent of exposure of hydrophobic patches and aggregate size are structural determinants of oligomer toxicity, with a high level of hydrophobicity and small size associated with the ability to cause cellular dysfunction[5,19,47–50]. We therefore probed these two parameters following the interaction of the $A\beta_{42}$ oligomers with trodusquemine. The degree of hydrophobicity of the oligomers was evaluated using the fluorescent dye 8-anilino-1-naphthalenesulfonate (ANS), which binds to hydrophobic patches on the surfaces of solvent-exposed protein aggregates[51]. A total of 15 μM ANS was added to 5 μM $A\beta_{42}$ oligomers (monomer equivalents) after incubation for 1 h at 20 °C in the absence or presence of between 10:1 and 1:10 molar ratios of $A\beta_{42}$-to-trodusquemine (Fig. 4a). We observed that the ANS fluorescence intensity increased in a well-defined dose-dependent manner, indicating that trodusquemine interacts with the $A\beta_{42}$ oligomers and that the solvent-exposed hydrophobicity of the oligomers increases with increasing levels of trodusquemine.

We next assessed the effect of trodusquemine on the size of the aggregates using static light scattering. Following the introduction of increasing concentrations of trodusquemine in the same ratios as those used in the ANS measurements, enhanced light scattering was observed with increasing concentrations of trodusquemine (Fig. 4b), providing further evidence of an interaction between trodusquemine and $A\beta_{42}$ oligomers. More-over, our results indicate that trodusquemine interacts more strongly with aggregated $A\beta_{42}$ (Fig. 4) than with the monomeric species (Supplementary Figure 3), and that this interaction changes the physicochemical properties of the aggregates. This phenomenon is in agreement with previous data reported in the literature for other oligomeric systems, where an increase in size was found to be concomitant with an increase in hydrophobicity[48,52]. The increases in size and hydrophobicity caused by trodusquemine were found to be significantly higher only at super-stoichiometric concentrations relative to the oligomers; however, sub-stoichiometric or equimolar concentrations were sufficient to impart a protective effect of

trodusquemine (Fig. 3). Indeed, at a 1:1 molar ratio, trodusquemine suppresses significantly the oligomer toxicity, but the increase in both size and hydrophobicity of the oligomers is small. It is unlikely, therefore, that the only mechanism of action through which trodusquemine reduces the toxicity of $A\beta_{42}$ oligomers is related to a substantial change of their hydrophobicity and size.

**Increased aggregation but reduced toxicity in AD worms.** To characterize the effects of trodusquemine in vivo, we made use of a transgenic *C. elegans* model of AD, in which human $A\beta_{42}$ is overexpressed in muscle tissue[53]. These animals, referred to hereafter as AD worms, have a phenotypic dysfunction in their ability to undergo body bends, commonly measured as the thrashing frequency or motility, in units of bends per minute (BPM), and the resultant reduced motility has been shown to be proportional to the levels of toxic species generated during the aggregation of $A\beta_{42}$ into fibrils[8]. We first sought to assess the levels of $A\beta_{42}$ aggregation by staining the worms with NIAD-4, a dye that is specific for amyloid fibrils[8], in AD worms with and without the administration of trodusquemine starting from the L4 larval stage of development prior to adulthood. Aggregation was observed in the AD worms by monitoring the level of fluorescence that results from the interaction of NIAD-4 with amyloid inclusions, and in the presence of trodusquemine, the levels of staining increased significantly and progressively (Fig. 5a). Indeed, at day 6 of adulthood (D6), 20 μM trodusquemine increased the measured levels of aggregates by more than six times in comparison to untreated AD worms.

At specific time points during their adult lifespan, the worms were monitored in a highly sensitive manner using the Wide Field-of-View Nematode Tracking Platform[22,54]. The overall fitness of each experimental group of worms was measured according to three characteristics after the administration of trodusquemine at the L4 stage of development, namely body bend frequency, speed of swimming and the extent of paralysis (Fig. 5b, Supplementary Figure 12). All these characteristics were moved significantly towards the behaviour of the control worms as a result of the administration of both 10 μM and 20 μM trodusquemine (Fig. 5b). To illustrate this point, the parameters were summed to generate a fitness score, which reflects the overall

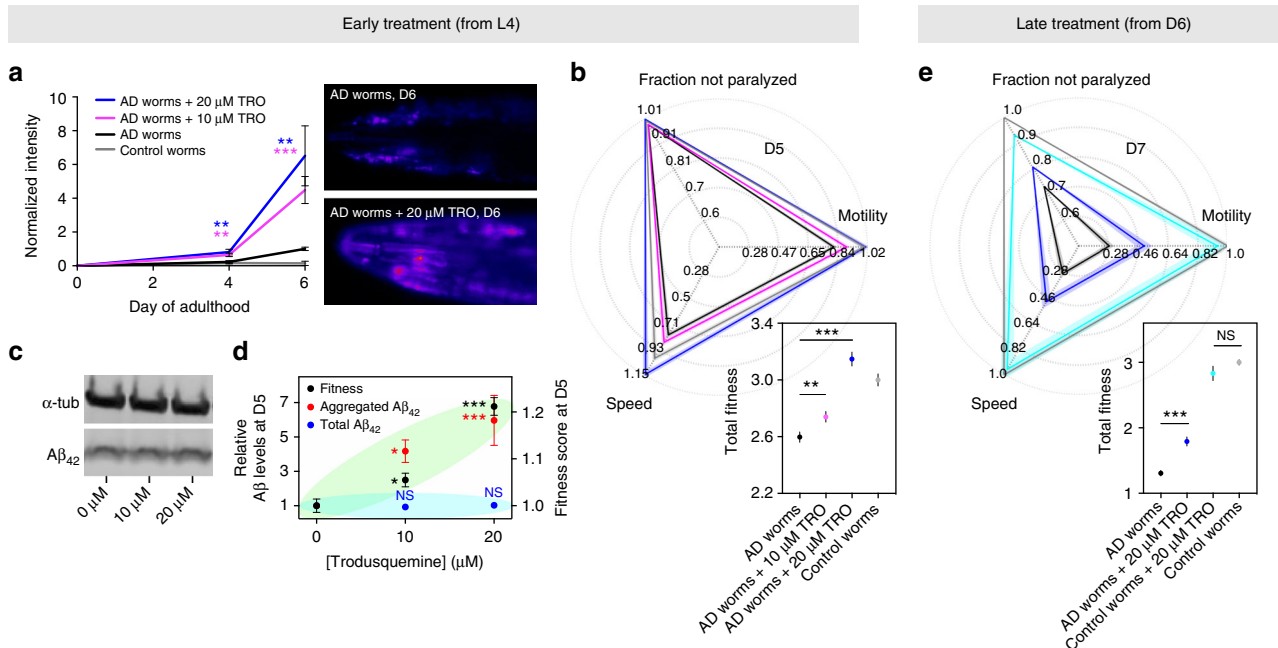

**Fig. 5** Increased aggregation, but reduced toxicity in a *C. elegans* model of AD. **a–d** Worms were treated at the L4 stage of development. **a** NIAD-4 staining of AD worms incubated in the absence (black) or presence of 10 μM (purple) and 20 μM (blue) doses of trodusquemine. Representative images at D6 are shown. \*\**P* < 0.01, \*\*\**P* < 0.001 by Student's *t*-test. Untreated control worms (grey) are shown for comparison. *N* = 20 per condition. **b** Motility, speed of swimming and the fraction of worms not paralyzed were monitored for AD worms incubated in the absence (*N* = 342) or presence of 10 μM (*N* = 352) and 20 μM (*N* = 309) doses of trodusquemine. These parameters were summed to generate fitness scores. \*\**P* < 0.01, \*\*\**P* < 0.001 by Student's *t*-test. Untreated control worms (*N* = 320) are shown for comparison. **c** Immunoblot measurements of the total Aβ levels in AD worms after treatment. Data shown are representative of duplicate blotting procedures. **d** Correlation between the concentration of trodusquemine administered and the increase in AD worm fitness (black, from **b**, \**P* < 0.05, \*\*\**P* < 0.001 by one-way ANOVA), the relative quantity of aggregated Aβ$_{42}$ (red, averages of D4 and D6 from **a**, \**P* < 0.05, \*\*\**P* < 0.001 by one-way ANOVA) and the total levels of Aβ$_{42}$ (blue, Supplementary Figure 13, NS: not significant by one-way ANOVA). All one-way ANOVA were followed by Bonferroni's multiple comparison test relative to untreated AD worms. Panels **b–d** represent D5 of adulthood. **e** AD worms were also treated at D6 with (*N* = 88) and without 20 μM TRO (*N* = 105) and monitored at D7 for the conditions described in **b**. Fitness scores were calculated as above. \*\*\**P* < 0.001 by Student's *t*-test. Untreated (*N* = 434) and treated (*N* = 84) control worms (cyan) are shown for comparison. NS: not significant by Student's *t*-test. All data represent mean ± s.e.m. (line thickness or error bar) with the *N*-values listed throughout. Aggregation and behavioural plots are representative of three biological replicates

animal health; trodusquemine promoted the recovery of worms when administered at the L4 stage of development (Supplementary Movies 1 and 2), and worms treated with 20 μM trodusquemine were shown to be approximately as healthy as wild-type control worms (Fig. 5b).

Treatment with trodusquemine did not alter the total levels of Aβ$_{42}$ in the AD worms (Fig. 5c, Supplementary Figure 13), which supports the conclusion that the beneficial effects of trodusquemine are not related to changes in Aβ$_{42}$ accumulation. We then considered the dependency of the fitness scores, the extent of aggregated Aβ$_{42}$ and the total levels of Aβ$_{42}$ on the concentrations of trodusquemine administered to the AD worms, for which a clear correlation between the extent of aggregation and worm health was observed (Fig. 5d), suggesting that these processes are related. As a control, trodusquemine was administered under the same conditions to wild-type worms (N2 strain) not expressing Aβ$_{42}$ (Supplementary Movies 3 and 4), and no improvement to body bend frequency or speed of swimming was observed. A minor decrease in paralysis was, however, observed (Supplementary Figure 14), suggesting that the increase in lifespan in the Aβ$_{42}$ expressing worms may include a modest contribution from changes in non-specific pathways regulating autophagy or other apparatuses of the proteostasis network. Indeed, trodusquemine has recently been identified as being capable of promoting regeneration in various tissues of zebrafish and mice[55] and longevity in wild type and Parkinson's disease worm models[23].

In an additional set of experiments, AD worms displaying a toxic phenotype (i.e. reduced body bend frequency) after incubation under standard conditions were treated at D6 with a 20 μM dose of trodusquemine to assess the ability of the molecule to stimulate the recovery of such worms. Indeed, the treatment-induced positive effects on the frequency of body bends, the speed of swimming and the extent of paralysis at late-stages in the worm lifespan (Fig. 5e). Overall, the fitness score of the treated AD worms was increased by 50% relative to untreated worms, as indicated by the average change in fitness across all time points measured in the lifespan of the worms (Supplementary Figure 12). As a control, no detectable enhancement of the fitness readouts was observed upon the treatment of wild-type worms (not expressing Aβ$_{42}$) with the addition of 20 μM trodusquemine at D6 of their lives (Fig. 5e).

Collectively, these results demonstrate that trodusquemine reduces the toxicity related to Aβ$_{42}$ aggregation in AD worms at the same time as it generates an increase in the rate of amyloid formation (Fig. 5). We showed previously that designed antibodies can cause comparable effects on toxicity in AD worms by reducing significantly the quantity of aggregated Aβ$_{42}$ when administered under similar early and late treatment paradigms[18]. The observed decreases in toxicity in both cases are highly likely to be related to changes in Aβ$_{42}$ oligomer behaviour, where the antibodies dramatically reduce the rate of their formation and trodusquemine both hastens their conversion to the mature

amyloid state and markedly attenuates their affinity for cell membranes.

## Discussion

The identification of oligomers as the species of $A\beta_{42}$ aggregates most capable of causing cellular toxicity has resulted in an increased focus on these aggregates as the key pathogenic agents in AD[1,5,7,17]. Such aggregates of $A\beta_{42}$ exist in a complex and dynamical heterogeneous mixture, and include metastable oligomers of various morphologies[2,17,31]. We have, therefore, sought to provide quantitative grounds to assess the nature and origin of the various toxic aggregates of $A\beta_{42}$ by using trodusquemine to modulate the behaviour of the aggregated species.

We first determined the rates of the various microscopic steps associated with the aggregation of $A\beta_{42}$ in the presence of trodusquemine. The results show that this compound enhances the overall process of aggregation, predominantly by increasing the rate of the monomer-dependent secondary nucleation. The mechanism by which the rate of secondary nucleation is increased in the presence of trodusquemine could be related to a change in the affinity of the monomers for fibrils after trodusquemine binds to their surfaces; indeed, aminosterols have been shown previously to bind amyloid fibrils[22,23]. It is also possible that an increase in the rate of conversion of pre-nucleation species into growth competent oligomers, or other steps in the nucleation process, may also play a role in the enhancement of aggregation. Spermine was observed to enhance the aggregation of $A\beta_{42}$ to a lesser extent than trodusquemine, and its accelerative effect was not significant under conditions of increased ionic strength. Trodusquemine, however, was observed to be effective at accelerating the aggregation reaction under conditions of moderate and high ionic strength, indicating that the covalent linkage of spermine to the sterol ring in trodusquemine is important in its ability to promote aggregation.

A decrease in the lengths of the fibrils in the presence of trodusquemine was also observed and found to be consistent with the expectations from the chemical kinetic data. In particular, increasing the rate of secondary nucleation would be predicted to shift the reactive flux of the aggregation reaction towards reduced fibril lengths, just as decreasing the rate of secondary nucleation with the molecular chaperone Brichos has previously been shown to increase fibril length[15]. We note that it has been proposed previously that enhancing the rate of aggregation alone has the potential to result in an overall reduction in toxicity by rapidly converting or consuming low molecular weight oligomers, into less toxic higher order species, ranging from high molecular weight aggregates to mature amyloid fibrils[10,19,56–59]. Indeed, our results show that the exogenous administration of trodusquemine reduces $A\beta_{42}$-mediated toxicity in a *C. elegans* model of AD in conjunction with an increase in the rate of in vivo amyloid accumulation, but without exerting any detectable changes in the total $A\beta_{42}$ levels.

An increase in the rate of aggregation that is linked to a decrease in toxicity has also been observed endogenously in *C. elegans* in long-lived *daf-2* mutants[60] and upon inhibition of the IGF-1 signaling pathway[61]. In contrast to the uncontrolled aggregation that is associated with numerous protein misfolding diseases, regulated aggregation has been shown to protect the cell through a variety of mechanisms during ageing and stress[62]. One such protective mechanism includes the utilization of specific highly evolved chaperones to sequester and convert the smallest, most cytotoxic oligomers into larger aggregates[19,47,63]. These strategies are prone to failure, however, as a consequence of the limited capacity of such systems in the presence of unexpectedly high levels of aggregation, for example, as a result of ageing or

during the progression of AD[47,64]. It is possible that trodusquemine promotes processes that sequester excess $A\beta_{42}$ aggregates, a feature which has been shown to be neuroprotective with molecular chaperones in *C. elegans* during ageing[60].

Confocal microscopy analysis of oligomers of $A\beta_{42}$ incubated with human neuroblastoma cells in the presence of trodusquemine showed that these oligomers have a diminished affinity for cell surfaces, a finding that correlates with markedly reduced levels of toxicity, as a result of the ability of trodusquemine to bind to cell membranes and displace $A\beta_{42}$ oligomers. In addition, a biophysical characterization of the oligomers showed that the degree of surface hydrophobicity was amplified upon their interaction with trodusquemine, and such changes in hydrophobic solvent exposure were concurrent with increases in aggregate size. The creation of a more hydrophobic intermediate aggregate may provide a molecular basis for the promotion of fibril formation[65], as solvent-exposed oligomers can grow to bury their hydrophobic surfaces within the core of a more ordered fibril with tighter packing[59]. The structurally similar aminosterol squalamine has been shown to interact with phospholipid bilayers[66], and squalamine and trodusquemine have been observed to displace oligomers of α-synuclein from the cellular membrane[22,23]. Consequently, the impact of the induced displacement of oligomeric aggregates from the cell surface is likely to be very significant, and in addition to play a key role in the observed suppression of $A\beta_{42}$ oligomer cytotoxicity.

On the basis of these findings, we suggest that reducing the toxicity of the oligomeric species by itself could represent a powerful therapeutic strategy. Indeed, we have observed previously that aminosterols can decrease the toxicity associated with protein aggregates, but in that case the rate of α-synuclein aggregate formation was also reduced[22,23]. In contrast, our present results show that trodusquemine enhances the rate of $A\beta_{42}$ aggregation in vitro and in a *C. elegans* model of AD, but markedly reduces the toxicity caused by aberrant $A\beta_{42}$ aggregates in neuronal cells and worms. While trodusquemine attenuates oligomer binding and toxicity, the increase in aggregation rate is also consistent with an enhanced conversion of oligomeric species to mature fibrils, a process that could increase the overall aggregation rate while reducing the concentration of intermediate oligomeric species, thus further reducing toxicity. The finding that trodusquemine suppresses the toxicity caused by $A\beta_{42}$ aggregation in a *C. elegans* model of AD when administered both prior to and after the development of an $A\beta_{42}$-mediated phenotype suggests that trodusquemine, and other molecules that reduce the toxicity of aggregated species, could be valuable lead compounds in the search for rational methods of treating AD.

## Methods

**Chemicals**. Trodusquemine was synthesized as a hydrochloride salt at a purity >97% as measured by mass spectrometry[23], stored as a lyophilized powder and solubilized in water to a final concentration of 10 mM[23]. Spermine (>97% purity, Sigma-Aldrich, MO, USA) was solubilized in water to a final concentration of 100 mM. Both molecules were stored in water at −20 °C until use.

**Preparation of $A\beta_{42}$ for chemical kinetics experiments**. The recombinant $A\beta_{42}$ peptide (MDAEFRHDSGY EVHHQKLVFF AEDVGSNKGA IIGLMVGGVV IA) was expressed in the *Escherichia coli* BL21 Gold (DE3) strain (Stratagene, CA, USA) and purified as described previously[9]. The purification procedure involved sonication of the *E. coli* cells, and a subsequent dissolution of the inclusion bodies in 8 M urea. Ion exchange chromatography was then performed on a diethylaminoethyl cellulose resin, and the protein collected was lyophilized. The lyophilized fractions were further purified using a Superdex 75 26/60 column (GE healthcare, IL, USA), and the eluates were analysed using SDS-polyacrylamide gel electrophoresis to detect the presence of the $A\beta_{42}$ peptide. The fractions containing the recombinant protein were collected, and subjected to lyophilization again.

Solutions of monomeric protein were prepared by purifying the lyophilized $A\beta_{42}$ peptide in 6 M GuHCl. The monomeric protein was purified by gel filtration in 20 mM sodium phosphate buffer, 200 µM EDTA, pH 8.0 using a Superdex

75 10/300 column (GE Healthcare) at a flow rate of 0.5 ml/min. In particular, EDTA was used to chelate any residual metal ions in the buffer, which are known to affect Aβ aggregation[67]. ThT was added from a 2 mM stock to give a final concentration of 20 μM. Experiments measuring the effect of spermine and trodusquemine on the aggregation reaction of Aβ$_{42}$ at different ionic strengths were performed in conditions of 5 mM sodium phosphate, 200 μM EDTA, pH 8.0 in the presence of either 75 or 150 mM NaCl. All samples were prepared in low-binding Eppendorf tubes and samples were analyzed in a 96-well half area, low-binding, clear-bottom PEG coated plate (Corning 3881).

For the seeded experiments, pre-formed fibrils were prepared just before the experiment. Kinetic experiments were set up just as above for a 4 μM Aβ$_{42}$ sample in 20 mM sodium phosphate, 200 μM EDTA, pH 8.0, and 20 μM ThT. The ThT fluorescence was monitored over time to ensure that the fibrils were formed. The samples were then collected from the wells into low-binding tubes.

**Kinetic analysis**. ThT fluorescence was monitored in triplicate per sample as measured using bottom-optics in a plate reader (Fluostar Omega or Fluostar Optima from BMG Labtech, Aylesbury, UK) with 440 and 480 nm excitation and emission filters, respectively. Samples were prepared on ice in the absence or presence of trodusquemine at 0.2, 0.4, and 2 μM. Aggregation was initiated by transferring the 96-well plate to the plate reader at 37 °C under quiescent conditions. The time evolution of the total fibril mass concentration, M(t), is described by the following integrated rate law[31]:

$$\frac{M(t)}{M(\infty)} = 1 - \left( \frac{B_+ + C_+}{B_+ + C_+ e^{\kappa t}} \frac{B_- + C_+ e^{\kappa t}}{B_- + C_+} \right)^{\frac{k_\infty^2}{\tilde{k}_\infty}} e^{-k_\infty t}, \qquad (2)$$

where the kinetic parameters $B_\pm$, $C_\pm$, $\kappa$, $k_\infty$ and $\tilde{k}_\infty$ are functions of the two combinations of the microscopic rate constants $k_+ k_2$ and $k_+ k_n$, where $k_n$, $k_+$ and $k_2$ are the primary nucleation, elongation and secondary nucleation rate constants, respectively. The perturbation induced by trodusquemine can then be resolved by fitting experimental data to the master equation to identify the alteration(s) of microscopic process(es), as described in the main text. In particular, we have performed the in vitro experiments under the buffer conditions described above, such that we are able to measure accurately and quantitatively the effects of trodusquemine in a well-characterized aggregation system of Aβ$_{42}$[9,15,18]. The fits were done using the AmyloFit platform[33], which uses a basin-hopping algorithm to find the best fit to the data. Half-time quantifications were similarly determined using AmyloFit[33].

**Atomic force microscopy (AFM)**. Solutions containing Aβ$_{42}$ fibrils were deposited on mica positively functionalized with (3-aminopropyl)triethoxysilane (APTES, Sigma-Aldrich, MO, USA) in the absence of ThT. The incubation time of 4 h was selected as this time was sufficient for all samples to enter the plateau phase of the chemical kinetics experiments. The mica substrate was positively functionalized prior to sample deposition by the incubation of a 10 μl drop of 0.05% (v/v) APTES in deionized water (Milli-Q, Merck Millipore, MA, USA) for 1 min at ambient temperature, rinsed with deionized water and then dried by the passage of a gentle flow of gaseous nitrogen[68]. AFM samples were prepared on the freshly functionalized MICA surfaces by the deposition of a 10 μL drop of protein (2 μM) for 2 min. Salts were washed away with water and the samples were stored in sealed containers until imaging was carried out using a JPK Nanowizard2 system (JPK Instruments, Berlin, Germany) operating in tapping mode with scan rates <0.5 Hz and a silicon tip with a 10 nm nominal radius (2 N m$^{-1}$, Micromasch, Wetzlar, Germany).

The interaction of the AFM tip with the sample causes changes in its motion. Whereas, the change in amplitude is related to sample morphology, the change in phase reflects the dissipated energy during the sample-tip interaction[35]. The measurement of each sample at low forces of interaction and constant phase change, and therefore constant tip-sample interaction, causes low sample deformation by the tip (<10%)[68], and enables us to compare consistently the morphology of different samples. We thus established standardized experimental scanning conditions and maintained a regime of phase change on the order of ≈Δ20° upon the interaction of the tip with the sample[35]. In all, 8–14 high resolution (1024 by 1024 pixels), phase-controlled images were acquired for each 4 h sample, for which one representative image is shown in Fig. 2. Three-dimensional maps were flattened using SPIP (Image Metrology, Hørsholm, Denmark) software, and lengths were determined by tracing along the median axis of each fibril. Heights were quantified by determining the cross-sectional diameter of each unique aggregate[35,68]. The sensitivity error for the AFM measurements of the height was calculated as the sum of the instrument electrical noise (<0.05 nm for the JPK system) and the sample roughness (<0.1 nm), and these errors were added to the standard error of the mean (SEM) to determine the total error as reported in the text. A two-sample t-test for means was carried out in OriginPro (OriginLab, MA, USA) with a test mean value greater than the sensitivity error.

**Transmission electron microscopy (TEM)**. Samples were prepared as described in the AFM experiments and deposited on a 400-mesh, 3-mm copper grid carbon support film (EM Resolutions Ltd., Sheffield, UK) and stained with 2% uranyl

acetate (w/v). Salts and excess uranyl acetate were washed by rinsing with deionized water (Milli-Q). Imaging was carried out on an FEI Tecnai G2 transmission electron microscope (Cambridge Advanced Imaging Centre, CAIC, University of Cambridge, UK), and the images were acquired using the SIS Megaview II Image Capture system (Olympus, Muenster, Germany). Fourteen to twenty images were acquired per 1 h and 4 h samples, for which one representative image is shown in Fig. 2, and the fibril widths were quantified in ImageJ (NIH, MD, USA) by tracing the cross-section perpendicular to the fibril axis. In the case of periodicity, the width of individual fibrils was measured at the crossover point (minimum width of the fibril) and between the crossovers (maximum width of the fibril), and both values were included in the final analysis. Fibrils were only quantified if they could be identified as discrete, individual aggregates (i.e. clustered or ambiguous fibrils were excluded). Images for quantification were acquired at ×14,500 magnification corresponding to 0.7 nm per pixel, and the total errors reported in the text were calculated as the sum of the error associated with the resolution and the SEM. A two-sample t-test for means was carried out in OriginPro with a test mean value greater than the sensitivity error.

**Nuclear magnetic resonance (NMR) spectroscopy**. Uniformly [15]N-labeled recombinantly expressed Aβ$_{42}$ (rPeptide, GA, USA) was prepared following the manufacturer's instructions and stored at −80 °C until use. Samples were prepared using 50 μM protein in the absence and presence of 50 μM trodusquemine (5 mM sodium phosphate, pH 7.5). While trodusquemine is highly soluble in water, its solubility in 5 mM sodium phosphate is limited above 50 μM. Eight scans were taken for each spectrum using a 500 MHz NMR (Bruker) at 5 °C, and the assignments were obtained from previous spectra[69].

**Preparation of oligomers**. Lyophilized Aβ$_{42}$ (Sigma-Aldrich, MO, USA) was dissolved in 100% hexafluoro-2-isopropanol (HFIP) to 1.0 mM and then the solvent was evaporated. Aβ-derived diffusible ligands (ADDLs) were prepared from Aβ$_{42}$ solutions according to Lambert's protocol[41] and analysed using western blotting and a dot blot assay to ensure they were closely similar to those previously reported[41]. Western blotting was performed by separating 2.25 μg of ADDLs on a 4–12% criterion XT Precast Bis-Tris gel probed with 1:800 mouse monoclonal 6E10 antibodies (803001, BioLegend, CA, USA) and 1:3000 peroxidase-conjugated anti-mouse secondary antibodies (AB6728, Abcam, Cambridge, UK). Dot blots were carried out by spotting 2 μl of monomers, ADDLs and fibrils (1 μg in monomer equivalents) and probing with 1:500 human anti-ADDLs therapeutic antibody (clone 19.3, TAB-0813CLV, Creative Biolabs, NY, USA) or with 1:800 mouse monoclonal 6E10 antibodies and then with 1:1000 peroxidase-conjugated anti-human secondary antibodies (A0170, Sigma-Aldrich, MO, USA) or with 1:3000 peroxidase-conjugated anti-mouse secondary antibodies.

**Neuroblastoma cell cultures**. Human SH-SY5Y neuroblastoma cells (Sigma-Aldrich, MO, USA, origin from A.T.C.C., VA, USA) were cultured in DMEM, F-12 HAM with 25 mM HEPES and NaHCO$_3$ (1:1) and supplemented with 10% FBS, 1 mM glutamine and 1.0% antibiotics. Cell cultures were maintained in a 5% CO$_2$ humidified atmosphere at 37 °C and grown until they reached 80% confluence for a maximum of 20 passages[46,70]. The cell line was authenticated by the European Collection of Authenticated Cell Cultures using short tandem repeat loci analyses and tested negative for mycoplasma contaminations.

**MTT reduction assay**. Cell viability was assessed by means of the 3-(4,5-dimethylthiazol-2-yl)-2,5-diphenyltetrazolium bromide (MTT) reduction assay[46]. Aβ$_{42}$ oligomers (1 μM monomer equivalents), prepared according to Lambert's protocol[41], were incubated with or without increasing concentrations (0.1, 0.33 and 1 μM) of trodusquemine for 1 h at 37 °C under shaking conditions, and then added to the cell culture medium of SH-SY5Y cells seeded in 96-well plates for 24 h. The Aβ$_{42}$-to-trodusquemine molar ratios used here were 10:1, 3:1, 1:1 for TRO. Samples were distributed throughout the mutliwell plate using a random allocation approach. Following 24 h, the cells were incubated with 0.5 mg/ml MTT at 37 °C for 4 h, and with cell lysis buffer (20% SDS, 50% N,N-dimethylformamide, pH 4.7) for 3 h. The absorbance values of blue formazan were determined at 590 nm. Cell viability was expressed as the percentage of MTT reduction in treated cells as compared to untreated cells. Pre-treatment experiments were carried out by incubating the cells for 15 min with 1 μM trodusquemine, after which time they were washed with PBS and subsequently exposed to oligomers for 24 h.

**Measurement of intracellular ROS**. Aβ$_{42}$ oligomers (1 μM monomer equivalents) were added to the cell culture medium of SH-SY5Y cells seeded on glass coverslips for 15 min, in the absence or presence of 1 μM trodusquemine. To detect intracellular ROS production, cells were then loaded with 10 μM 2′,7′-dichlorodihydrofluorescein diacetate (CM-H$_2$DCFDA, Life Technologies, CA, USA)[70]. Pre-treatment experiments were carried out by incubating the cells for 15 min with 1 μM trodusquemine, after which time they were washed with PBS and subsequently exposed to oligomers for 15 min. Cells were labeled with the CM-H$_2$DCFDA probe in the last 10 min of treatment, and the resulting fluorescence was analyzed by a TCS SP5 scanning confocal microscopy system (Leica Microsystems, Mannheim, Germany) equipped with an argon laser source, using the 488 nm excitation line. A

series of 1.0 μm thick optical sections (1024 × 1024 pixels) were taken through the cells for each sample using a Leica Plan Apo ×63 oil immersion objective (Leica Microsystems, Mannheim, Germany) and then projected as a single composite image by superimposition. The confocal microscope was set at optimal acquisition conditions, e.g., pinhole diameters, detector gain and laser powers. Settings were maintained constant for each analysis.

**Oligomer binding to the cellular membrane.** SH-SY5Y cells were seeded on glass coverslips and treated for 15 min with $A\beta_{42}$ oligomers (1 μM monomer equivalents) in the absence or presence of 1 μM trodusquemine. Pre-treatment experiments were carried out by incubating the cells for 15 min with 1 μM trodusquemine, after which time they were washed with PBS and subsequently exposed to oligomers for 15 min. After incubation, the cells were washed with PBS and counterstained with 5.0 μg/ml Alexa Fluor 633-conjugated wheat germ agglutinin (Life Technologies, CA, USA)[22]. After washing with PBS, the cells were fixed in 2% (w/v) buffered paraformaldehyde for 10 min at room temperature (RT, 20 °C). The presence of oligomers was detected with 1:800 diluted mouse monoclonal 6E10 anti-Aβ antibodies (BioLegend, CA, USA) and subsequently with 1:1000 diluted Alexa Fluor 488-conjugated anti-mouse secondary antibodies (Life Technologies, CA, USA). To detect only the oligomers bound to the cell surface, the cellular membrane was not permeabilized at this stage, thus preventing antibody internalization. The median planes of the cells were also analyzed to exclude the possibility that the green fluorescence arises from the cytosol. Fluorescence emission was detected after double excitation at 488 nm and 633 nm by the scanning confocal microscopy system described above, and three apical sections were projected as a single composite image by superimposition. Binding histograms were created using ImageJ (NIH, MD, USA) and JACOP plugin (rsb.info. nih.gov) software.

**Analysis of oligomer co-localization with lysosomes.** SH-SY5Y cells were treated for 15 min with 1 μM ADDLs formed from HiLyte™ Fluor 647-labeled Aβ42 (AnaSpec, CA, USA), washed with PBS and labelled with 50 nM of the LysoTracker® Green DND-26 probe (Thermo Fisher Scientific, Waltham, MA, USA) for 30 min. Untreated or chloroquine-treated (50 μM, 4 h) cells were used as negative and positive controls. The lysosomes and the oligomers were detected by using double excitation at 488 nm and 647 nm.

**ANS binding measurements.** $A\beta_{42}$ oligomers (5 μM monomer equivalents) were incubated in the absence or presence of 5, 12.5, 25, 37.5 and 50 μM concentrations of trodusquemine in 20 mM Tris, 100 mM NaCl for 1 h at 20 °C. 8-anilino-1-naphthalenesulfonate (ANS, Sigma-Aldrich, MO, USA) was then added to a final concentration of 15 μM. Fluorescence emission spectra were recorded using a plate reader (BMG Labtech, Aylesbury, UK) with excitation at 380 nm. Representative data are shown in Fig. 4 for three-independent experiments.

**Light scattering.** Samples were prepared as described above, but in the absence of ANS. Static light scattering measurements were performed at 25 °C using the Malvern Zetasizer Nano S instrument (Malvern, Worcestershire, UK) with fixed parameters (attenuator 10, cell position 4.65 mm), equipped with a Peltier temperature controller. A low volume (100 μl) disposable cuvette was used (BRAND, Wertheim, Germany).

**C. elegans strains.** All strains of C. elegans were acquired from the Caenorhabditis Genetics Center (CGC), which is funded by National Institutes of Health Office of Research Infrastructure Programs (P40 OD010440). Two strains were utilized for these experiments. The temperature sensitive human amyloid beta expressing strain dvIs100 [unc-54p::A-beta-1–42::unc-54 3′-UTR+mtl-2p::GFP] (GMC101) was used, in which mtl-2p::GFP causes intestinal GFP expression and unc-54p::A-beta-1–42 expresses the human full-length $A\beta_{42}$ in the muscle cells of the body wall. Raising the temperature above 20 °C at the L4 or adult stage causes paralysis due to $A\beta_{42}$ aggregation in the body wall muscle. The N2 strain was used for wild-type worms.

**C. elegans media.** Standard conditions were used for the propagation of C. elegans[71]. Briefly, the animals were synchronized by hypochlorite bleaching and subsequently hatched overnight in M9 buffer (3 g/l $KH_2PO_4$, 6 g/l $Na_2HPO_4$, 5 g/l NaCl, 1 μM $MgSO_4$), after which they were cultured at 20 °C on nematode growth medium (NGM) ($CaCl_2$ 1 mM, $MgSO_4$ 1 mM, cholesterol 5 μg/ml, 250 μM $KH_2PO_4$ pH 6, agar 17 g/L, NaCl 3 g/l, casein 7.5 g/l) plates seeded with the E. coli strain OP50. Saturated cultures of OP50 were grown by inoculating 50 mL of LB medium (tryptone 10 g/l, NaCl 10 g/l, yeast extract 5 g/l) with OP50 and incubating the culture for 16 h at 37 °C. NGM plates were seeded with bacteria by adding 350 μl of saturated OP50 to each plate and leaving the plates at 20 °C for 2–3 days. On day 3 after synchronization, the animals were placed on NGM plates containing 5-fluoro-2′-deoxyuridine (FUdR, Sigma-Aldrich, MO, USA) (75 μM) to inhibit the growth of offspring and stored for the duration of their lifespan at 24 °C.

**Trodusquemine administration to C. elegans.** NGM plates containing FUdR were seeded with 2.2 ml aliquots of trodusquemine at the appropriate concentration or water as the vehicle and dried in a laminar flow hood at RT. Worms were transferred by random allocation to plates coated with trodusquemine or water at the L4 stage of development or D6. Worms treated with trodusquemine at D6 were maintained on FUdR plates from the L4 stage of development at 24 °C, as described above.

**Automated C. elegans phenotypic assay.** At the times indicated and for each condition, worms were washed off the plates with M9 buffer and spread over an OP50 unseeded 9 cm NGM plate in a final volume of 5 mL, after which their movements were recorded at 20 fps using a recently developed microscopic procedure for 60 s[22,54]. The sample size was determined based on previous studies[8,22]. Videos were analyzed in a consistent manner to track worm motility (bends per minute), speed of swimming and the extent of paralysis; paralyzed worms were defined by the code as being those which move less than 5 bends per minute and with a swimming speed of less than 1 mm/min. Further details regarding the hardware of the tracker and its associated code are available from [54] and from the authors. All data shown are representative of three experiments from individual developmentally synchronized populations of worms.

**NIAD-4 staining of live C. elegans.** Live animals were incubated with NIAD-4 over a range of concentrations and times, and it was empirically determined that incubation of living animals with 1 μM NIAD-4 (0.1% DMSO in M9 buffer) for 4 h at room temperature gave robust and reproducible staining. After staining, animals were allowed to recover on NGM plates for about 24 h to allow destaining via normal metabolism. Stained animals were mounted on 2% agarose pads containing 40 mM $NaN_3$ as anaesthetic on glass microscope slides for imaging[8]. Images were captured with a Zeiss Axio Observer A1 fluorescence microscope (Carl Zeiss Microscopy GmbH, Jena, Germany) with a ×20 objective and a 49004 ET-CY3/TRITC filter (Chroma Technology Corp, VT, USA). Fluorescence intensity was calculated using ImageJ software (NIH, MD, USA) and then normalized as the corrected total cell fluorescence. Only the head region was considered because of the high background signal in the guts. All experiments were carried out in triplicate, and the data from one representative experiment are shown. Twenty animals were analyzed per condition. The representative images shown in Fig. 5a have been uniformly modified by the 'fire' filter in ImageJ to facilitate the visualization of the aggregates.

**C. elegans immunoblots.** Animals were prepared and treated with trodusquemine from the L4 stage of development as described above. At D5 of adulthood, ~6000 worms per condition were collected from FUdR plates and samples were extracted by sonication in 8 M urea, 2% SDS, 50 mM DTT, 50 mM Tris, pH 8.0 with 1X proteinase inhibitor cocktail (Roche, Mannheim, Germany). Lysates of each sample were combined with NuPAGE LDS sample buffer (1X) and NuPAGE sample reducing agent (1X) (Life Technologies, CA, USA) and heated for 10 min at 70 °C. Samples were separated using a NuPAGE Novex 4–12% Bis-Tris Protein gel (Life Technologies, CA, USA). Gels were incubated for 10 min at RT in 10% ethanol prior to transferring to a nitrocellulose membrane (iBlot Dry Blotting System, Life Technologies, CA, USA). Membranes were cut and incubated at 4 °C overnight with 1:1000 monoclonal anti-α-tubulin antibodies (clone B-5-1-2, T5168, Sigma-Aldrich, MO, USA) and the bottom section was incubated with 1:1500 monoclonal anti-Aβ antibodies (clone W0-2, MABN10, Merck Millipore, MA, USA). 1:5000 Alexa 488-conjugated secondary antibody (anti-mouse, Life Technologies, CA, USA) was incubated for 1 h at RT, and membranes were measured for fluorescence using a Typhoon Trio Imager (GE Healthcare, IL, USA). Monomeric $A\beta_{42}$ was prepared as described above and used as a positive control. The lysate of control worms not expressing $A\beta_{42}$ was used as a negative control. Lysates were run in triplicate and band intensities were quantified in ImageJ (NIH, Bethesda, MD) using gel analysis. Data shown are representative of duplicate experiments that gave consistent results. Uncropped blots are shown in Supplementary Figure 13.

**Statistical analysis.** Except where otherwise stated, comparisons between the different groups were performed by one-way ANOVA followed by Bonferroni's post comparison test and the unpaired, two-tailed Student's t-test, and all statistical tests were performed in GraphPad Prism 7.0 (CA, USA). $P < 0.05$ was accepted as statistically significant.

## Data availability

Data supporting the findings of this manuscript are available from the corresponding authors upon reasonable request.

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

## Acknowledgements

The authors thank Pietro Sormanni for key contributions to the *C. elegans* data analysis and graphical representation, and Ewa Klimont and Swapan Preet for protein expression and purification. This work was supported by the Cambridge Centre for Misfolding Diseases (R.L., S.C., F.S.R., M.P., G.T.H., G.M., B.M., J.H., T.C.T.M., P.K.C., M.A., S.T.C., N.F., C.K.X., N.D.K., J.R.K., T.P.J.K., M.V. and C.M.D.), the UK Biotechnology and Biochemical Sciences Research Council (M.V. and C.M.D.), the Wellcome Trust (T.P.J.K., M.V. and C.M.D.), the Frances and Augustus Newman Foundation (T.P.J.K.), the Regione Toscana—FAS Salute, project SUPREMAL (R.C., C.C. and F.C.), the European Research Council (S.L.), Darwin College Cambridge (F.S.R.), Sidney Sussex College Cambridge (G.M.), Peterhouse Cambridge (T.C.T.M.), the Swiss National Science Foundation (T.C.T.M.), a Herchel Smith Research Studentship (C.K.X.), the Agency for Science, Technology, and Research, Singapore (S.C.), the Gates Cambridge Trust (R.L. and G.T.H.) and St. John's College Cambridge (R.L.). The NMR facility (Department of Chemistry, University of Cambridge) is supported, in part, by an EPSRC Core Capability grant (EP/K039520/1).

## Author contributions

S.C., G.M., J.H., S.L., T.C.T.M. and R.L. performed the in vitro aggregation experiments and the associated data analysis. R.L. and F.S.R. performed the AFM measurements and data analysis. R.C. performed the cell experiments and their analysis. R.L. and B.M. performed the biophysical characterisation of Aβ42 oligomers with input from C.K.X. and N.D.K. R.L., M.P., S.T.C., N.W.F. and P.K.C. performed the *C. elegans* measurements and data analysis. G.T.H. and M.A. performed the NMR measurements and analysis. R.L. and J.R.K. performed the TEM measurements and data analysis. R.L., S.C., F.S.R., M.P., R.C., G.T.H., G.M., B.M., J.H., J.R.K., C.C., M.Z., S.L., T.P.J.K., F.C., M.V. and C.M.D. were involved in the design of the study. R.L., F.C., M.V. and C.M.D. wrote the manuscript. All authors were involved in the analysis of data and editing of the paper.

## Additional information

**Competing interests:** The authors declare the following competing interests: M.Z. is one of the inventors in a patent for the use of trodusquemine in the treatment of Parkinson's disease. C.M.D., M.V., S.I.A.C., T.P.J.K., J.H. and S.L. are co-founders of Wren Therapeutics Limited, which is independently pursuing inhibitors of protein misfolding and aggregation. The remaining authors declare no competing interests.

