## [Peer Review File · Nature Communications]

Reviewers' Comments:

Reviewer #1:

Remarks to the Author:

This manuscript makes a good case that trodusquemine alters the kinetics of Abeta in vitro aggregation and can impact the toxicity of Abeta in SH-SY5Y cells and a *C. elegans* model. The weakness of the study is a lack of clarity about whether the biological effects of trodusquemine are due primarily to its enhancement of Abeta amyloid formation or its ability to block Abeta interactions with membranes, or something else (e.g., induction of autophagy?). This issue could be potentially resolved by additional experiments, specifically:

1) I was surprised the authors did not assay toxicity in SH-SY5Y cells pre-treated with trodusquemine and then exposed to Abeta oligomers (prepared without trodusquemine). If this treatment gives similar protection as observed when the Abeta oligomers are formed in the presence of trodusquemine, it would imply that the effects of the compound are due to membrane interactions rather than trodusquemine effects on Abeta aggregation.

2) The authors need to assay Abeta levels and distribution in the *C. elegans* model after trodusquemine treatment. It is completely plausible that the reduced toxicity is due to reduced levels of Abeta accumulation (i.e., due to increased autophagy or reduced transgene expression) rather than alteration of Abeta structures or interactions with membranes. In particular, an immunoblot of treated and untreated worm lysates could determine both overall Abeta levels and the distribution of stable oligomers.

3) It has been known for 15 years that induction of Abeta expression in *C. elegans* can lead to paralysis independent of detectable amyloid formation (see Drake et al, 2003, Fig 2). If trodusquemine treatment reduced paralysis rates in the inducible *C. elegans* models and also led to the accumulation of detectable amyloid it would significantly strengthen the case that the equilibrium shift to amyloidic Abeta was driving the reduced toxicity.

Reviewer #2:

Remarks to the Author:

This paper reports on a small molecule that is capable of converting toxic oligomeric Abeta into less toxic fibrillary aggregates. It follows a recent paradigm shift that oligomeric Abeta is the likely seminal etiological agent of AD, whereas its fibrillary form may, in fact, have protective features. Although this concept has been around for a few years now, there are only very few examples of successful implementation of that approach, and these findings will be therefore of very high interest and relevance to the field. Literature citation is appropriate and data generally of high quality, with analysis supporting the claims for the most part. I would recommend publication after some minor changes to the manuscript:

1. The authors talk in the introduction about Abeta misfolding, which I find a little misleading, since Abeta is an inherently disordered peptide and, as such, does not have a defined native structure. I know that the term "misfolding" has been commonly applied to this IDP, but it may be more appropriate to talk about conformational sampling, as discussed by Granata Vendruscolo et al in *Sci. Rep.*

2. The punctal staining that the authors see in Figure 3C may be more consistent with lysosomal intracellular co-localization of Abeta than membrane binding. The authors should compare their findings with those reported in the paper by X. Hu, J.-M. Lee et al (PNAS 2009), especially the images shown in Figure 2, since their staining pattern is very similar. I would like to see additional lysosomal staining experiments, so as to ensure that their Abeta is not, in fact, lysosomal.

3. The authors talk about a "well-defined dose dependence", but in Figure 3A, the reduction in Abeta cytotoxicity upon addition of 1,5 or 10 equivalents of TRO all appear to be within error. This needs to be re-phrased so that claims made in the text are consistent with the data presented.

Other than that I believe that this is a very fine contribution, and would be happy to recommend it for publication, once the concerns above have been addressed.

Reviewer #3:

Remarks to the Author:

Limbocker et al. measured the aggregation rate of (2 μM) A β 42 using Thioflavin T fluorescence, at increasing concentrations of trodusquemine (TRO) from 0.2- 2 μM . They also examined aggregation in the presence of spermine, at ionic strengths of 75 and 150 mM. To better understand the kinetics of aggregation they added either 5% or 25% fibril seeds at the beginning of aggregation, and analyzed the time to reach half of the maximum fluorescence. Next they examined the interaction of monomeric A β 42 with TRO using NMR spectroscopy. They also analyzed the morphology of fibrils formed in the absence and presence of TRO by TEM and AFM. Moving to cell culture experiments, they tested TRO's ability to prevent the toxicity from synthetic A β 42 ADDLs on human neuroblastoma cells, using MTT reduction, and by imaging ROS production, as well as association with cellular membranes. They also characterized the changes in ADDL ANS binding as a function of increasing amounts of TRO. Finally, they test TRO in a *C. elegans* model of AD to determine if there are changes in aggregate intensity, and worm motility, under two paradigms; treatment starting from L4 (before aggregates have formed) or from D6 (after aggregates have formed).

They find that A β 42 aggregated faster (shorter half time) in the presence of TRO. Spermine also increases the aggregation but not as much as TRO, further they find that 150 mM NaCl could reverse spermine's accelerating effects, but not TROs, indicating that the interaction of TRO with A β 42 does not primarily rely on ionic interactions. They observed that in the presence of TRO the fibrils formed at 4 hours were shorter, higher and wider than those normally formed, which is consistent with the results from the kinetics study. In cell culture they found that TRO was able to reverse the toxicity of ADDLs, as measured by MTT reduction and ROS production. They also showed a 75% reduction in the 6E10 positive A β 42 that interacts stably with SH-SY5Y membranes. They observed an increase in ANS fluorescence intensity with increasing amounts of TRO, indicating that ADDL hydrophobic exposure is increased by the addition of TRO, as well as an upward trend in size according to light scattering. In *C. elegans* the authors find that treatment with TRO increases the intensity of the aggregates of A β , and improves their motility (fitness) back to control levels, suggesting that TRO promotes the formation of large aggregates in agreement with the ThT data.

This work builds on prior studies from the group on the very similar squalamine inhibitor, on the aggregation of α -synuclein. The work is thorough, carefully performed and thoughtfully considered.

Major comments:

A hypothesis should be put forward for how TRO could increase secondary nucleation, and/or how it binds to oligomeric structures. Does TRO bind to fibrils and somehow facilitate the binding of additional monomers?

No examination of the structures formed by following the ADDL production procedure of Lambert, 2008 is provided. We do not know anything about these other than their ability to reduce MTT reduction by 23% compared to untreated cells, and to "stick to" the membranes of cells as measured by the 6E10 antibody. Because much of the text emphasizes the importance of oligomers and their toxicity, understanding something about the structure of the oligomers under study seems to be critical to their conclusions.

Minor comments:

The graphing of the data for the *C. elegans* studies is problematic. Very similar measures were done in both the L4 and D6 treatment paradigms, yet the data is presented in very different ways. For example: Why is Fig. 4b graphed on a semicircle, and not in the same manner as Fig. 4e? Graphing them in the same way (as Fig. 4e) would make for a more straight-forward comparison. Why are Fig. 4c and 4f plotted differently? Again, this makes comparison of the data more difficult. The bar graph seems to be the appropriate manner to plot this data, they should both be presented like this. (Treatment from L4 would merely have D3, D7 and D11.)

Additional points:

1. Provide justification as to why is pH 8 (and not physiological pH 7.4) used throughout, and why is EDTA included the buffers for the in vitro experiments?
2. For aggregation kinetics in the presence of TRO, a reduction by 2x is already seen at a 5:1 ratio, not only at 1:1, as stated on line 127.
3. When the experiments with spermine are first introduced on line 132, justification should be provided for doing this experiment.
4. So as to see the structures present at t=0 by AFM, the z-scale (color scale) could be adjusted to less than 8 nm on Fig. 2a.
5. On line 223 the sentence should read "...the effects of sample deformation...".
6. On lines 245-247, the following statement should be clarified: "We therefore conclude that the presence of trodusquemine accelerates the supramolecular organization of the individual fibrils into larger species, a result that is consistent with the increased final ThT signal observed from the kinetic experiments."
7. On line 259, it should read "reduced by a factor of ≈ 0.4 (Fig. 2e)" not "reduced by ≈ 0.4 (Fig. 2e)".
8. With regard to the MTT experiments, the authors should comment on the reduction of viability with the highest TRO concentration. Is the 96% of untreated cells significant? It seems that only A β 42 with TRO at increasing ratios were examined statistically.
9. In the paragraph on the top of page 11, the probe used for ROS detection (CM-H2DCFDA) should be mentioned in the text.
10. What is the source for the ANS?
11. Which points in the light scattering data (Fig. 3e) with TRO are significantly different from ADDLs alone or from each other?
12. For the treatment of *C. elegans* why were the concentrations of 10 and 20 μ M TRO used? Is the A β 42 level in the worms known? Does this correspond to a 1:1 molar ratio or something else?
13. With regard to the *C. elegans* study, the authors report there is no difference at for the control worms treated with TRO at D7 or D12, but it appears that there IS a significant difference at D10, was this examined (statistically), if this is so, what does it indicate?
14. Error bars are missing on Fig. 4e, part two "Fraction not paralyzed".
15. The paragraph that begins on line 477 should be written more concisely. (It currently repeats the observations for the interaction of squalamine with α -synuclein multiple times.)

The manuscript by Dobson and co-workers entitled "Trodesquamine enhances secondary nucleation in Abeta42 aggregation but suppresses its associated toxicity by displacing Abeta42 oligomers from cell membranes" is very well written and expertly executed and merits serious consideration for publication.

This paper provides kinetic and microscopic evidence that Trodesquamine utilizes surface binding to preformed fibrils of Abeta42 to rapidly accelerate secondary nucleation to generate short fibrils. Toxicity in an SH-SY5Y cancer cell model read out by MTT reduction and improved fitness in an Abeta42 muscle secretion C. elegans model suggests that as the Abeta42 monomer equivalents : Trodesquamine concentration ratio becomes 1:1, Trodesquamine more strongly interacts with Abeta42 oligomers changing the physical properties of the aggregates and blocking SH-SY5Y plasma membrane binding, which could explain the amelioration in oligomer proteotoxicity.

It is interesting that a molecule that enhances aggregation by a monomer dependent secondary nucleation mechanism. In the neuroblastoma and C.elegans models this acceleration of aggregation coupled by a coating of the oligomers seems to be a dual mechanism to reduce proteotoxicity.

I suggest that the authors consider the possibility that hastened secondary nucleation is also consuming faster low MW oligomers that are hard to quantify but are responsible for the degenerative phenotypes in humans. NOT for this paper, but in the future the authors should consider whether two color fluorescence correlation spectroscopy or microscopy could be used to show or disprove that short oligomers are substrates for secondary nucleation on larger oligomers of Abeta42.

The data in this paper are compelling, and suggest that a human clinical study may be the future rather than murine model studies that have proven to be lousy predictors of drug efficacy in humans. Trodesquamine, a natural product, is a Protein Tyrosine Phosphatase 1B inhibitor in phase II human clinical trials, as inhibition of Protein Tyrosine Phosphatase 1B is thought to be beneficial to combat Diabetes and Obesity. It might be good for the authors to add a paragraph to the discussion about how good the blood brain permeability of Trodesquamine is and what its safety profile looks like to do, as a prelude to how practical it is to think about clinical trials with this compound.

This is a very nice paper and it should be published without delay, as the experiments are well conceived and executed and the results are compelling and the well written description is logical and interesting.

Reviewers' comments:

Reviewer #1 (Remarks to the Author):

This manuscript makes a good case that trodusquemine alters the kinetics of Abeta in vitro aggregation and can impact the toxicity of Abeta in SH-SY5Y cells and a *C. elegans* model. The weakness of the study is a lack of clarity about whether the biological effects of trodusquemine are due primarily to its enhancement of Abeta amyloid formation or its ability to block Abeta interactions with membranes, or something else (e.g., induction of autophagy?). This issue could be potentially resolved by additional experiments, specifically:

- 1) I was surprised the authors did not assay toxicity in SH-SY5Y cells pre-treated with trodusquemine and then exposed to Abeta oligomers (prepared without trodusquemine). If this treatment gives similar protection as observed when the Abeta oligomers are formed in the presence of trodusquemine, it would imply that the effects of the compound are due to membrane interactions rather than trodusquemine effects on Abeta aggregation.
- 2) The authors need to assay Abeta levels and distribution in the *C. elegans* model after trodusquemine treatment. It is completely plausible that the reduced toxicity is due to reduced levels of Abeta accumulation (i.e., due to increased autophagy or reduced transgene expression) rather than alteration of Abeta structures or interactions with membranes. In particular, an immunoblot of treated and untreated worm lysates could determine both overall Abeta levels and the distribution of stable oligomers.
- 3) It has been known for 15 years that induction of Abeta expression in *C. elegans* can lead to paralysis independent of detectable amyloid formation (see Drake et al, 2003, Fig 2). If trodusquemine treatment reduced paralysis rates in the inducible *C. elegans* models and also led to the accumulation of detectable amyloid it would significantly strengthen the case that the equilibrium shift to amyloidic Abeta was driving the reduced toxicity.

Reviewer #2 (Remarks to the Author):

This paper reports on a small molecule that is capable of converting toxic oligomeric Abeta into less toxic fibrillary aggregates. It follows a recent paradigm shift that oligomeric Abeta is the likely seminal etiological agent of AD, whereas its fibrillary form may, in fact, have protective features. Although this concept has

been around for a few years now, there are only very few examples of successful implementation of that approach, and these findings will be therefore of very high interest and relevance to the field. Literature citation is appropriate and data generally of high quality, with analysis supporting the claims for the most part. I would recommend publication after some minor changes to the manuscript:

1. The authors talk in the introduction about Abeta misfolding, which I find a little misleading, since Abeta is an inherently disordered peptide and, as such, does not have a defined native structure. I know that the term "misfolding" has been commonly applied to this IDP, but it may be more appropriate to talk about conformational sampling, as discussed by Granata Vendruscolo et al in Sci. Rep.
2. The punctal staining that the authors see in Figure 3C may be more consistent with lysosomal intracellular co-localization of Abeta than membrane binding. The authors should compare their findings with those reported in the paper by X. Hu, J.-M. Lee et al (PNAS 2009), especially the images shown in Figure 2, since their staining pattern is very similar. I would like to see additional lysosomal staining experiments, so as to ensure that their Abeta is not, in fact, lysosomal.
3. The authors talk about a "well-defined dose dependence", but in Figure 3A, the reduction in Abeta cytotoxicity upon addition of 1,5 or 10 equivalents of TRO all appear to be within error. This needs to be re-phrased so that claims made in the text are consistent with the data presented.

Other than that I believe that this is a very fine contribution, and would be happy to recommend it for publication, once the concerns above have been addressed.

Reviewer #3 (Remarks to the Author):

Limbocker et al. measured the aggregation rate of (2 μ M) A β 42 using Thioflavin T fluorescence, at increasing concentrations of trodusquemine (TRO) from 0.2- 2 μ M. They also examined aggregation in the presence of spermine, at ionic strengths of 75 and 150 mM. To better understand the kinetics of aggregation they added either 5% or 25% fibril seeds at the beginning of aggregation, and analyzed the time to reach half of the maximum fluorescence. Next they examined the interaction of monomeric A β 42 with TRO using NMR spectroscopy. They also analyzed the morphology of fibrils formed in the absence and presence of TRO by TEM and AFM. Moving to cell culture experiments, they tested TRO's ability to prevent the toxicity from synthetic A β 42 ADDLs on human neuroblastoma cells, using MTT reduction, and by imaging ROS production, as well as association with cellular membranes. They also characterized the changes in ADDL ANS binding as a function of increasing amounts of TRO. Finally, they test TRO in a *C. elegans*

model of AD to determine if there are changes in aggregate intensity, and worm motility, under two paradigms; treatment starting from L4 (before aggregates have formed) or from D6 (after aggregates have formed).

They find that A β 42 aggregated faster (shorter half time) in the presence of TRO. Spermine also increases the aggregation but not as much as TRO, further they find that 150 mM NaCl could reverse spermine's accelerating effects, but not TROs, indicating that the interaction of TRO with A β 42 does not primarily rely on ionic interactions. They observed that in the presence of TRO the fibrils formed at 4 hours were shorter, higher and wider than those normally formed, which is consistent with the results from the kinetics study. In cell culture they found that TRO was able to reverse the toxicity of ADDLs, as measured by MTT reduction and ROS production. They also showed a 75% reduction in the 6E10 positive A β 42 that interacts stably with SH-SY5Y membranes. They observed an increase in ANS fluorescence intensity with increasing amounts of TRO, indicating that ADDL hydrophobic exposure is increased by the addition of TRO, as well as an upward trend in size according to light scattering. In *C. elegans* the authors find that treatment with TRO increases the intensity of the aggregates of A β , and improves their motility (fitness) back to control levels, suggesting that TRO promotes the formation of large aggregates in agreement with the ThT data.

This work builds on prior studies from the group on the very similar squalamine inhibitor, on the aggregation of α -synuclein. The work is thorough, carefully performed and thoughtfully considered.

Major comments:

A hypothesis should be put forward for how TRO could increase secondary nucleation, and/or how it binds to oligomeric structures. Does TRO bind to fibrils and somehow facilitate the binding of additional monomers?

No examination of the structures formed by following the ADDL production procedure of Lambert, 2008 is provided. We do not know anything about these other than their ability to reduce MTT reduction by 23% compared to untreated cells, and to "stick to" the membranes of cells as measured by the 6E10 antibody. Because much of the text emphasizes the importance of oligomers and their toxicity, understanding something about the structure of the oligomers under study seems to be critical to their conclusions.

Minor comments:

The graphing of the data for the *C. elegans* studies is problematic. Very similar measures were done in both the L4 and D6 treatment paradigms, yet the data is presented in very different ways. For example: Why is Fig. 4b graphed on a semicircle, and not in the same manner as Fig. 4e? Graphing them in the same way (as Fig. 4e) would make for a more straight-forward comparison. Why are Fig.

4c and 4f plotted differently? Again, this makes comparison of the data more difficult. The bar graph seems to be the appropriate manner to plot this data, they should both be presented like this. (Treatment from L4 would merely have D3, D7 and D11.)

Additional points:

1. Provide justification as to why is pH 8 (and not physiological pH 7.4) used throughout, and why is EDTA included the buffers for the in vitro experiments?
2. For aggregation kinetics in the presence of TRO, a reduction by 2x is already seen at a 5:1 ratio, not only at 1:1, as stated on line 127.
3. When the experiments with spermine are first introduced on line 132, justification should be provided for doing this experiment.
4. So as to see the structures present at t=0 by AFM, the z-scale (color scale) could be adjusted to less than 8 nm on Fig. 2a.
5. On line 223 the sentence should read "...the effects of sample deformation...".
6. On lines 245-247, the following statement should be clarified: "We therefore conclude that the presence of trodusquemine accelerates the supramolecular organization of the individual fibrils into larger species, a result that is consistent with the increased final ThT signal observed from the kinetic experiments."
7. On line 259, it should read "reduced by a factor of ≈ 0.4 (Fig. 2e)" not "reduced by ≈ 0.4 (Fig. 2e)".
8. With regard to the MTT experiments, the authors should comment on the reduction of viability with the highest TRO concentration. Is the 96% of untreated cells significant? It seems that only A β 42 with TRO at increasing ratios were examined statistically.
9. In the paragraph on the top of page 11, the probe used for ROS detection (CM-H2DCFDA) should be mentioned in the text.
10. What is the source for the ANS?
11. Which points in the light scattering data (Fig. 3e) with TRO are significantly different from ADDLs alone or from each other?
12. For the treatment of *C. elegans* why were the concentrations of 10 and 20 μ M TRO used? Is the A β 42 level in the worms known? Does this correspond to a 1:1 molar ratio or something else?
13. With regard to the *C. elegans* study, the authors report there is no difference at for the control worms treated with TRO at D7 or D12, but it appears that there IS a significant difference at D10, was this examined (statistically), if this is so, what does it indicate?
14. Error bars are missing on Fig. 4e, part two "Fraction not paralyzed".
15. The paragraph that begins on line 477 should be written more concisely. (It currently repeats the observations for the interaction of squalamine with α -synuclein multiple times.)

Reviewer #4 (Remarks to the Author):

The manuscript by Dobson and co-workers entitled "Trodesquamine enhances secondary nucleation in Aβ42 aggregation but suppresses its associated toxicity by displacing Aβ42 oligomers from cell membranes" is very well written and expertly executed and merits serious consideration for publication.

This paper provides kinetic and microscopic evidence that Trodesquamine utilizes surface binding to preformed fibrils of Aβ42 to rapidly accelerate secondary nucleation to generate short fibrils. Toxicity in an SH-SY5Y cancer cell model read out by MTT reduction and improved fitness in an Aβ42 muscle secretion *C. elegans* model suggests that as the Aβ42 monomer equivalents : Trodesquamine concentration ratio becomes 1:1, Trodesquamine more strongly interacts with Aβ42 oligomers changing the physical properties of the aggregates and blocking SH-SY5Y plasma membrane binding, which could explain the amelioration in oligomer proteotoxicity.

It is interesting that a molecule that enhances aggregation by a monomer dependent secondary nucleation mechanism. In the neuroblastoma and *C. elegans* models this acceleration of aggregation coupled by a coating of the oligomers seems to be a dual mechanism to reduce proteotoxicity.

I suggest that the authors consider the possibility that hastened secondary nucleation is also consuming faster low MW oligomers that are hard to quantify but are responsible for the degenerative phenotypes in humans. NOT for this paper, but in the future the authors should consider whether two color fluorescence correlation spectroscopy or microscopy could be used to show or disprove that short oligomers are substrates for secondary nucleation on larger oligomers of Aβ42.

The data in this paper are compelling, and suggest that a human clinical study may be the future rather than murine model studies that have proven to be low predictors of drug efficacy in humans. Trodesquamine, a natural product, is a Protein Tyrosine Phosphatase 1B inhibitor in phase II human clinical trials, as inhibition of Protein Tyrosine Phosphatase 1B is thought to be beneficial to combat Diabetes and Obesity. It might be good for the authors to add a paragraph to the discussion about how good the blood brain permeability of Trodesquamine is and what its safety profile looks like to do, as a prelude to how practical it is to think about clinical trials with this compound.

This is a very nice paper and it should be published without delay, as the experiments are well conceived and executed and the results are compelling and the well written description is logical and interesting.

Author reply (point by point basis)

We are most grateful to Nature Communications, its editors and the reviewers for taking the time and effort to carefully review our manuscript. It is clear that addressing their responses, the manuscript has improved in a significant way. After carrying out the experiments suggested by the reviewers and amending the manuscript accordingly, we sincerely hope that this revision exceeds the high standards for publication in Nature Communications.

Herein, reviewer comments are shown in black and responded to in blue. All corresponding changes to the manuscript and supplementary information have been highlighted.

Reviewers' comments:

Reviewer #1 (Remarks to the Author):

1) I was surprised the authors did not assay toxicity in SH-SY5Y cells pre-treated with trodusquemine and then exposed to Abeta oligomers (prepared without trodusquemine). If this treatment gives similar protection as observed when the Abeta oligomers are formed in the presence of trodusquemine, it would imply that the effects of the compound are due to membrane interactions rather than trodusquemine effects on Abeta aggregation.

We thank Reviewer 1 for his/her careful assessment of our work. We agree that these measurements would strength the conclusions regarding the ability of trodusquemine to displace $A\beta_{42}$ oligomers. We therefore pre-treated SH-SY5Y cells with trodusquemine followed by a thorough washing step with PBS and the subsequent addition of $A\beta_{42}$ oligomers formed in the absence of trodusquemine, for which we observed that the molecule at an equimolar concentration to the oligomers maintains its potent ability to displace the toxic species, attenuate ROS production, and increase cellular viability. These results are described on page 12 and as red bars in **Fig. 3a-c**. Indeed, these results demonstrate clearly that the compound modifies the cell membrane and protects it from deleterious interactions with $A\beta_{42}$ oligomers, even without free molecule in solution with the oligomers.

2) The authors need to assay Abeta levels and distribution in the C. elegans model after trodusquemine treatment. It is completely plausible that the reduced toxicity is due to reduced levels of Abeta accumulation (i.e., due to increased autophagy or reduced transgene expression) rather than alteration of Abeta structures or interactions with membranes. In particular, an immunoblot of treated and

untreated worm lysates could determine both overall Abeta levels and the distribution of stable oligomers.

3) It has been known for 15 years that induction of Abeta expression in *C. elegans* can lead to paralysis independent of detectable amyloid formation (see Drake et al, 2003, Fig 2). If trodusquemine treatment reduced paralysis rates in the inducible *C. elegans* models and also led to the accumulation of detectable amyloid it would significantly strengthen the case that the equilibrium shift to amyloidic Abeta was driving the reduced toxicity.

(2 & 3) The reviewer sensibly notes that alternative pathways, in particular those related to protein expression and aggregate clearance, could contribute to the observed suppression of toxicity upon trodusquemine administration to *C. elegans* expressing human A β ₄₂. To assess the extent of A β ₄₂ aggregation in the worms, we chose previously to use the commonly employed dye NIAD-4, which binds to mature amyloid species in the worm (Habchi et al., *Sci. Adv.*, 2016; Aprile et al., *Sci. Adv.*, 2017). The staining showed that trodusquemine potentiates the formation of insoluble A β ₄₂ inclusions *in vivo* over time (**Fig. 4a**). These results indicate that amyloid aggregate accumulation is significantly stimulated in the worm by trodusquemine in tandem with an increase in worm health.

As far the accumulation of total A β ₄₂ is concerned, the reviewer is correct: we did not previously explore the total accumulation of A β ₄₂ in the AD worms with and without trodusquemine treatment. We therefore carried out the suggested immunoblots. First, we screened the untreated and 20 μ M treated worms at day 5 of adulthood to confirm the beneficial effects of trodusquemine in AD worms prior to initiating western blotting procedures. Indeed, in good agreement with our previous measurements, we observed that the molecule increased AD worm fitness significantly ($p < 0.001$, Student's t-test) and by approximately 32% in the samples prepared for western blotting (please see below; data are shown relative to the untreated group).

We then sonicated extensively the worm lysates in lysis buffer and carried out western blotting, which indicates that treatment with trodusquemine does not alter the total accumulation of $A\beta_{42}$ in the AD worm. The observed decrease in toxicity is therefore not likely to be related to alterations in transgene expression or autophagy activity. A representative western blot is shown in **Fig. 4c**. Lysates were analyzed in triplicate (**Supplementary Fig. 13**) to produce the quantification reported in **Fig. 4d**. Blotting procedures were carried out twice with identical results. These results are described on page 15.

We propose that the ability of trodusquemine to enhance aggregation *in vitro* and *in vivo* is likely to be linked to its functionality in binding membranes and suppressing the toxicity of its related pre-fibrillar aggregated species. It is further likely that the biological effect of the molecule is a function of both processes, where displacement protects the cell from oligomers and enhanced aggregation converts toxic oligomers to relatively inert higher order aggregates, as highlighted by Reviewer 4, and that these processes cooperate in the worms to reduce toxicity.

Reviewer #2 (Remarks to the Author):

1. The authors talk in the introduction about Abeta misfolding, which I find a little misleading, since Abeta is an inherently disordered peptide and, as such, does not have a defined native structure. I know that the term "misfolding" has been commonly applied to this IDP, but it may be more appropriate to talk about conformational sampling, as discussed by Granata Vendruscolo et al in Sci. Rep.

We have removed the term “misfolding” to clarify the text, with thanks to the reviewer.

2. The punctal staining that the authors see in Figure 3C may be more consistent with lysosomal intracellular co-localization of Abeta than membrane binding. The authors should compare their findings with those reported in the paper by X. Hu, J.-M. Lee et al (PNAS 2009), especially the images shown in Figure 2, since their staining pattern is very similar. I would like to see additional lysosomal staining experiments, so as to ensure that their Abeta is not, in fact, lysosomal.

To address this issue, we have carried out an additional experiment using the LysoTracker kit to monitor lysosomal activity in the cells in the presence of labeled A β ₄₂ oligomers. Our results show that the oligomers not only exert a minimal effect on lysosome activity, but also do not colocalize significantly with lysosomes. This result has been added in a new figure (**Supplementary Fig. 10**) and is described in the main text on page 12.

In addition, for the binding experiments carried out herein and previously (Evangelisti et al., *Sci. Rep.*, 2016; Perni et al., *PNAS*, 2017), we sought only to detect oligomers bound to the surface of the cellular membrane. In addition to monitoring the apical planes of the cells, to prevent antibody internalization, the cells were not permeabilized. Indeed, the antibodies used to identify the oligomers cannot pass the cellular membrane. It has, however, been demonstrated previously that intracellular oligomers can be observed in the medial planes of cells treated with oligomers under specific conditions (Pensalfini et al., *Neurobiol. Aging*, 2011). We therefore carried out an additional analysis of our existing data for cells treated with A β ₄₂ oligomers in the absence and presence of trodusquemine, and observed that, in our experiments, negligible fluorescence signal corresponding to intracellular oligomers could be detected both in the absence and presence of trodusquemine. These results have also been added in a new figure (**Supplementary Fig. 11**) and described in the main text on page 12, and further indicate that the green fluorescence that we observed in cells treated with A β ₄₂ oligomers arises only from the oligomers bound to the plasma membrane and not from those internalized.

Collectively, we conclude that it is unlikely that the A β ₄₂ is lysosomal, and the changes reported **Fig. 3c** can be attributed to alterations in membrane binding induced by trodusquemine.

3. The authors talk about a "well-defined dose dependence", but in Figure 3A, the reduction in Abeta cytotoxicity upon addition of 1,5 or 10 equivalents of TRO all appear to be within error. This needs to be re-phrased so that claims made in the

text are consistent with the data presented.

Amended on page 11, with thanks to the reviewer.

Reviewer #3 (Remarks to the Author):

Major comments:

A hypothesis should be put forward for how TRO could increase secondary nucleation, and/or how it binds to oligomeric structures. Does TRO bind to fibrils and somehow facilitate the binding of additional monomers?

The mechanism by which secondary nucleation is stimulated could be related to a change in the affinity of the monomer for the fibril after trodusquemine binds to its surface, as suggested by Reviewer 3. Indeed, aminosterols have been demonstrated previously to bind amyloid fibrils (Perni et al., *PNAS*, 2017; Perni et al., *ACS. Chem. Biol.*, 2018). It is furthermore possible that potentiated oligomer conversion or other processes may also play a role in the enhancement of aggregation. We have updated the Discussion on page 16 to include these possibilities.

While the present study details in full the ability of trodusquemine to displace toxic A β ₄₂ oligomers from the cell membrane, elucidating the precise mechanism by which trodusquemine binds to oligomeric species and impacts their physicochemical properties is the focus of future research.

No examination of the structures formed by following the ADDL production procedure of Lambert, 2008 is provided. We do not know anything about these other than their ability to reduce MTT reduction by 23% compared to untreated cells, and to "stick to" the membranes of cells as measured by the 6E10 antibody. Because much of the text emphasizes the importance of oligomers and their toxicity, understanding something about the structure of the oligomers under study seems to be critical to their conclusions.

The reviewer is correct: we had produced, used and shown experiments with ADDLs without showing any evidence that our samples contained the required species. Synthetic ADDLs have been studied with SDS-PAGE and were shown to contain 3mer to 24 mer (Lambert et al. 2001 *J. Neurochem.*; Klein 2002 *Neurochem. Int.*), with 12mer as the most prominent species (Gong et al. 2002 *PNAS*). They have also been studied with conformation-sensitive polyclonal M94 antiserum (Gong et al. 2003 *PNAS*) and with conformation-sensitive monoclonal NU-1 and NU-4 antibodies (Lambert et al., *J. Neurochem.*, 2007). To our knowledge a structural characterisation using other techniques has not been attempted by Bill Klein's group.

We have therefore chosen to use SDS-PAGE coupled with western blotting and dot-blot analyses using the only conformation-sensitive anti-ADDLs antibody that is available on the market. The results are in good agreement with the above-mentioned studies and are shown in the new **Supplementary Fig. 9** and briefly mentioned in the main text on page 11.

Minor comments:

The graphing of the data for the *C. elegans* studies is problematic. Very similar measures were done in both the L4 and D6 treatment paradigms, yet the data is presented in very different ways. For example: Why is Fig. 4b graphed on a semicircle, and not in the same manner as Fig. 4e? Graphing them in the same way (as Fig. 4e) would make for a more straight-forward comparison. Why are Fig. 4c and 4f plotted differently? Again, this makes comparison of the data more difficult. The bar graph seems to be the appropriate manner to plot this data, they should both be presented like this. (Treatment from L4 would merely have D3, D7 and D11.)

We have revised **Fig. 4** to enhance its clarity and to include the quantifications of total A β_{42} in the absence and presence of trodusquemine. In particular, we have added the speed of swimming for the worms as an additional metric of worm health (Hahm et al., *Nat. Commun.*, 2015 and Aprile et al., *Sci. Adv.*, 2017). Consistent with our previous findings, the increase in both the motility and the speed of swimming show that the AD worm healthspan is increased in the presence of trodusquemine, as exemplified by the total fitness scores. The effects of trodusquemine on AD worm health are comparable to our previous study that reduced the toxicity caused by A β_{42} aggregation in these animals by treatment with designed antibodies that inhibit the rate of amyloid formation *in vitro* and *in vivo* (Aprile et al., *Sci. Adv.*, 2017). We have added this comparison with a corresponding explanation to the results section on page 16.

We wish to include representative videos which demonstrate visually the effects of trodusquemine treatment in AD and control worms at D5 of adulthood (**Supplementary Videos 1-4**) to aid in the conceptualization of the behavioral plots shown in **Fig. 4**. We anticipate that this analysis will enable a more direct comparison of the effects of the molecule between the different treatment paradigms, and we thank the reviewer for their suggestions to improve the clarity of the results.

Additional points:

1. Provide justification as to why is pH 8 (and not physiological pH 7.4) used throughout, and why is EDTA included the buffers for the *in vitro* experiments?

The aggregation of A β ₄₂ is highly sensitive and influenced by extrinsic factors, such as ionic strength and pH (Meisl et al., *Sci. Rep.*, 2016 & Meisl et al., *Chem. Sci.*, 2017). A β ₄₂ in the buffer conditions that we have used is a well-characterised system (Cohen et al., *PNAS*, 2013), through which perturbations caused by small molecules, antibodies, and chaperones have been quantitatively determined (Habchi et al., *PNAS*, 2016; Aprile et al., *Sci. Adv.*, 2017; Cohen et al., *Nat. Struct. Mol. Biol.*, 2015). Thus, we sought to perform our *in vitro* experiments at pH 8 to describe quantitatively and accurately the effect that TRO has on the aggregation of A β ₄₂. This point has been added on page 20. Further, we also show that at higher ionic strength, where buffer conditions are more physiologically relevant, we observe that TRO is still able to accelerate the aggregation of A β (**Supplementary Fig. 1**).

In the aggregation of A β ₄₂, a pure peptide is purified, which is necessary to achieve reproducible kinetic data. EDTA is used in the buffers to chelate metal ions, which are known to affect the aggregation of amyloid beta (Abelein et al., *PNAS*, 2015). We have revised the manuscript on pages 19-20 to reflect this point.

2. For aggregation kinetics in the presence of TRO, a reduction by 2x is already seen at a 5:1 ratio, not only at 1:1, as stated on line 127.

Amended on page 5, with thanks to the reviewer.

3. When the experiments with spermine are first introduced on line 132, justification should be provided for doing this experiment.

We have added a justification on page 5, as requested by the reviewer.

4. So as to see the structures present at t=0 by AFM, the z-scale (color scale) could be adjusted to less than 8 nm on Fig. 2a.

As our analysis concerns only final fibrillar products, Figure 2a was placed on the same scale as Figure 2b to emphasize the absence of higher order aggregates at 0h of aggregation. While we suggest that this is the clearest representation of the data, we agree that the images at t = 0 h should be shown more explicitly (now added as **Supplementary Fig. 5**). Indeed, fibrils were not observed at the initiation stage (t = 0 h) of the aggregation reaction.

5. On line 223 the sentence should read "...the effects of sample deformation...".

Amended on page 8 with thanks to the reviewer.

6. On lines 245-247, the following statement should be clarified: "We therefore conclude that the presence of trodusquemine accelerates the supramolecular organization of the individual fibrils into larger species, a result that is consistent with the increased final ThT signal observed from the kinetic experiments."

We agree with the reviewer that this statement is not clear and perhaps speculative. Indeed, as stated in our previous sentence (lines 241-245 of the first submitted version), the AFM and TEM imaging shows with clarity that the incubation of A β ₄₂ in the presence of trodusquemine resulted in the formation of fibrils characterized by larger cross-sectional dimensions, as observed by the increased average height ($p < 0.001$, Fig. 2f) and width ($p < 0.001$, Fig. 2g), in comparison to fibrils formed in the absence of the molecule. However, it is not clear if this is due to an acceleration of the supramolecular organization of the individual fibrils or alternative pathways. We have therefore removed the statement altogether on page 9 and left the previous one to end the paragraph, as it pertains more to the experimental observations. We have also removed the corresponding text in the discussion related to this point.

7. On line 259, it should read "reduced by a factor of ≈ 0.4 (Fig. 2e)" not "reduced by ≈ 0.4 (Fig. 2e)".

Amended on page 9 with thanks to the reviewer.

8. With regard to the MTT experiments, the authors should comment on the reduction of viability with the highest TRO concentration. Is the 96% of untreated cells significant? It seems that only A β ₄₂ with TRO at increasing ratios were examined statistically.

The difference between untreated cells (100%) and the highest concentration of TRO in the absence of oligomers (97% of untreated cells) is not statistically significant, as has been clarified in the text on page 11. The reason why the two bars of oligomers + Tro 1:1 have the ° symbols of significance, whereas the bar for Tro has not any ° symbol, is because the symbols refer to cells treated with oligomers and untreated cells, respectively, as explained in the legend of **Fig. 3**.

9. In the paragraph on the top of page 11, the probe used for ROS detection (CM-H2DCFDA) should be mentioned in the text.

Amended on page 11 with thanks to the reviewer.

10. What is the source for the ANS?

The source (Sigma-Aldrich, MO, USA) has been added to the Methods section on page 24.

11. Which points in the light scattering data (Fig. 3e) with TRO are significantly different from ADDLs alone or from each other?

We have now performed an unpaired, two-tailed Student's t-test for each concentration of trodusquemine relative to oligomers in the absence of the molecule (**Fig. 3e**). This analysis indicates that oligomers incubated in the presence of 3-fold and 10-fold excesses of trodusquemine are significantly larger than oligomers alone ($p < 0.01$ and $p < 0.05$, respectively), which is in agreement with our previous conclusions regarding the effects of trodusquemine on oligomer size. An explanation of this analysis has been added to the Methods section on page 25. We thank the reviewer for highlighting the lack of a statistical analysis for this data in the first submitted version.

12. For the treatment of *C. elegans* why were the concentrations of 10 and 20 μM TRO used? Is the $\text{A}\beta_{42}$ level in the worms known? Does this correspond to a 1:1 molar ratio or something else?

These concentrations were initially tested based on past drug discovery work using *C. elegans* models of AD and PD. We have observed previously that the structurally similar aminosterol squalamine was admissible to Parkinson's disease model *C. elegans* and their controls in the range of 0-50 μM when administered at the L4 stage of development (Perni et al., *PNAS*, 2017). The treatment schedule used herein was influenced by this study, as it was observed that administration of squalamine at the L1 stage of development was toxic to juvenile worms (Perni et al., *PNAS*, 2017). Similarly, it has been observed that the small molecule bexarotene was able to inhibit $\text{A}\beta_{42}$ aggregation and reduce its toxicity in AD worms, with a maximal effect observed at a concentration of 10 μM when delivered at the L1 and L4 stages of development (Habchi et al., *Sci. Adv.*, 2016). From these two studies, we therefore initially elected to test concentrations of trodusquemine of 10 μM and 20 μM administered from the L4 stage, which was confirmed to be an effective treatment protocol in subsequent experiments. As to our knowledge the precise levels of $\text{A}\beta_{42}$ in the worm are not well understood, it is not possible to extract a molar ratio of the molecule to the total amount of $\text{A}\beta_{42}$.

13. With regard to the *C. elegans* study, the authors report there is no difference at for the control worms treated with TRO at D7 or D12, but it appears that there IS a significant difference at D10, was this examined (statistically), if this is so, what does it indicate?

The reviewer identifies that there is an apparent decrease in control worm health at D10 of adulthood when trodusquemine was administered under the late treatment conditions (**Fig. 4e** in the first submitted version), which was not previously examined statistically.

The data support our original claim that “no detectable enhancement to either fitness readout was observed upon the treatment of wild-type worms (not expressing $A\beta_{42}$) with the addition of 20 μ M trodusquemine at day 6 of their lives (**Fig. 4e**).” In the screening procedures herein, worms were monitored by collecting the animals from two plates in order to account for the heterogeneity that often exists within a worm population (Perni, *J. Neurosci. Methods*, 2018; Lublin and Link, *Drug Discov. Today Technol.*, 2013) and to minimize differences that can arise from sampling. It is likely that the difference at D10 is a result of sampling, as suggested by the unchanged control worm health upon trodusquemine treatment when measured at D7 and D12 of adulthood from the experiment. Nonetheless, our original conclusion that trodusquemine does not benefit wild-type worms is supported by the data, as was also confirmed in our biological replicates.

14. Error bars are missing on Fig. 4e, part two “Fraction not paralyzed”.

In Fig. 4e error bars are indicated by line thickness and refer to SEM, as reported in the legend. For the parameter “fraction not paralysed” the code used herein outputs a single value without error bar for the fraction of worms in each video moving below 5 bends per minute as “paralyzed.” We have calculated the “fraction not paralyzed” as 1-(paralyzed fraction), and we plotted this number in the various worm experiments to represent the number of worms that are moving.

15. The paragraph that begins on line 477 should be written more concisely. (It currently repeats the observations for the interaction of squalamine with α -synuclein multiple times.)

We have revised the text accordingly on page 18, with thanks to the reviewer for identifying this redundancy in the text.

Reviewer #4 (Remarks to the Author):

I suggest that the authors consider the possibility that hastened secondary nucleation is also consuming faster low MW oligomers that are hard to quantify but are responsible for the degenerative phenotypes in humans. NOT for this paper, but in the future the authors should consider whether two color fluorescence correlation spectroscopy or microscopy could be use to show or disprove that short oligomers are substrates for secondary nucleation on larger oligomers of Abeta42.

Studying the templating of short oligomers on larger ones as a model for secondary nucleation is a very interesting idea that we will consider for future studies. Regarding the point on the consumption of low-molecular weight oligomers, we have updated the Results section on page 16 and the Discussion section on page 17 to mention more explicitly this concept. Our new measurements showing that the total A β levels are unchanged by trodusquemine treatment in the AD worms, coupled with the increased rate of aggregation, are consistent with this conclusion and suggest that the decrease in toxicity is mediated, at least in part, through the promotion of the mature amyloid state by the molecule.

The data in this paper are compelling, and suggest that a human clinical study may be the future rather than murine model studies that have proven to be lousy predictors of drug efficacy in humans. Trodusquemine, a natural product, is a Protein Tyrosine Phosphatase 1B inhibitor in phase II human clinical trials, as inhibition of Protein Tyrosine Phosphatase 1B is thought to be beneficial to combat Diabetes and Obesity. It might be good for the authors to add a paragraph to the discussion about how good the blood brain permeability of Trodusquemine is and what its safety profile looks like to do, as a prelude to how practical it is to think about clinical trials with this compound.

We have expanded the Introduction section on page 4 to include more explicitly the points raised by the Reviewer regarding the safety, admissibility and blood-brain barrier permeability of trodusquemine to humans, in addition to its ongoing clinical trials. In consideration of the word limit, we have not expanded our previous remarks on the prior investigations of the aminosterols in the Discussion section. We thank the reviewer for these important suggestions.

Reviewers' Comments:

Reviewer #1:

Remarks to the Author:

The authors have undertaken additional experiments that have addressed my concerns and significantly improved the manuscript.

Reviewer #2:

Remarks to the Author:

The authors addressed my comments, and I am happy to recommend the paper for publication

Reviewer #3:

Remarks to the Author:

The authors have greatly improved the manuscript by addressing the comments of the four reviewers.

The experiments the authors performed in response to the reviews have strengthened their evidence, and support the claims in the revised manuscript.

I can now fully support the publication of this revised version of the manuscript.

REVIEWERS' COMMENTS:

Reviewer #1 (Remarks to the Author):

The authors have undertaken additional experiments that have addressed my concerns and significantly improved the manuscript.

Reviewer #2 (Remarks to the Author):

The authors addressed my comments, and I am happy to recommend the paper for publication

Reviewer #3 (Remarks to the Author):

The authors have greatly improved the manuscript by addressing the comments of the four reviewers. The experiments the authors performed in response to the reviews have strengthened their evidence, and support the claims in the revised manuscript.

I can now fully support the publication of this revised version of the manuscript.